# The influence of dataset homology and a rigorous evaluation strategy on protein secondary structure prediction

Teng-Ruei Chen[1,2☯], Chia-Hua Lo[3☯], Sheng-Hung Juan[1], Wei-Cheng Lo[1,2,3,4,5]*

1 Institute of Bioinformatics and Systems Biology, National Chiao Tung University, Hsinchu, Taiwan, 2 Institute of Bioinformatics and Systems Biology, National Yang Ming Chiao Tung University, Hsinchu, Taiwan, 3 Department of Biological Science and Technology, National Chiao Tung University, Hsinchu, Taiwan, 4 Department of Biological Science and Technology, National Yang Ming Chiao Tung University, Hsinchu, Taiwan, 5 The Center for Bioinformatics Research, National Yang Ming Chiao Tung University, Hsinchu, Taiwan

☯ These authors contributed equally to this work.
* WadeLo@nctu.edu.tw

**Data Availability Statement:** All relevant data are within the paper and its Supporting Information files.

**Funding:** This work was funded by the Ministry of Science and Technology (MOST), Taiwan (https://

## Abstract

The secondary structure prediction (SSP) of proteins has long been an essential structural biology technique with various applications. Despite its vital role in many research and industrial fields, in recent years, as the accuracy of state-of-the-art secondary structure predictors approaches the theoretical upper limit, SSP has been considered no longer challenging or too challenging to make advances. With the belief that the substantial improvement of SSP will move forward many fields depending on it, we conducted this study, which focused on three issues that have not been noticed or thoroughly examined yet but may have affected the reliability of the evaluation of previous SSP algorithms. These issues are all about the sequence homology between or within the developmental and evaluation datasets. We thus designed many different homology layouts of datasets to train and evaluate SSP prediction models. Multiple repeats were performed in each experiment by random sampling. The conclusions obtained with small experimental datasets were verified with large-scale datasets using state-of-the-art SSP algorithms. Very different from the long-established assumption, we discover that the sequence homology between query datasets for training, testing, and independent tests exerts little influence on SSP accuracy. Besides, the sequence homology redundancy between or within most datasets would make the accuracy of an SSP algorithm overestimated, while the redundancy within the reference dataset for extracting predictive features would make the accuracy underestimated. Since the overestimating effects are more significant than the underestimating effect, the accuracy of some SSP methods might have been overestimated. Based on the discoveries, we propose a rigorous procedure for developing SSP algorithms and making reliable evaluations, hoping to bring substantial improvements to future SSP methods and benefit all research and application fields relying on accurate prediction of protein secondary structures.

www.most.gov.tw/?l=en) with grant number NSC 101-2311-B-009-006-MY2 to WCL. The funders had no role in study design, data collection and analysis, decision to publish, or preparation of the manuscript.

**Competing interests:** The authors have declared that no competing interests exist.

## Introduction

The secondary structure prediction (SSP) of proteins is a technique to predict the backbone conformation of proteins based only on amino acid sequences. Although SSP has many applications, in our latest research, we found some potential problems in the development and evaluation process of previous SSP algorithms that might have made the accuracy misestimated. We suppose that if there can be a proper standard procedure for producing the experimental datasets and an accuracy verification scheme more precise than the currently used, they may help develop truly accurate SSP algorithms and move forward all related fields. By investigating the influence of the sequence redundancy of developmental datasets on SSP accuracy, this work aims to program a sophisticated SSP dataset preparation procedure and a rigorous evaluation strategy to prevent over/underestimating an SSP method during development.

SSP is an important basis for the prediction of protein tertiary structure [1–3] and is a crucial step in many fields, inclusive of functional prediction of proteins [4, 5], epitope prediction for antibodies [6–8], identification of disease-causing mutations or genetic variations [9–11], prediction of local properties of residues [12–14], discrimination of structured from intrinsically disordered protein regions [15–18], improvement of protein sequence alignment [19, 20], and template search or model refinement for protein structure modelling [21–23].

According to the secondary structure alphabet applied, SSP can be classified into three-state (Q3) and eight-state (Q8) predictions. Three-state predictors describe a protein conformation as helixes, strands, and coils/loops. Eight-state predictors use the 8 secondary structure elements (SSE) described by DSSP [24]. Therefore, the accuracy of SSP is also measured in two ways, the Q3 and Q8 accuracy. In the 1970s, the Q3 accuracy reached ~60% by analyzing amino acid propensities or the physicochemical properties of adjacent residues [25]. Subsequently, machine learning became the dominant SSP approach and raised the Q3 accuracy to ~65% in the late 1980s [26]. In 1999, PSIPRED [27], a neural network machine-learning method, first utilized the position-specific scoring matrices (PSSM) generated by PSI-BLAST [28] to be the main feature set and made Q3 achieve a new high, 76.5%. SSpro8 then pushed the Q8 accuracy to 62.6% also based on the PSSM [29]. After that, using the PSSM feature set became the mainstream in SSP. Nowadays, state-of-the-art algorithms, such as RaptorX, SpineX, Scorpion, Spider2/3, DeepCNF, MUFOLD-SS, NetSurfP-2, and Porter 5 [30–38], all used PSSM, and their Q3 and Q8 accuracy approximately fell in 81–85% and 71–75%, respectively.

To use PSSM as the feature set to construct an SSP model, the general scheme applied by current methods requires a "reference dataset" for homology search (usually the UniRef90 [39]), a query dataset for training, and another query dataset for testing. The reference dataset is also termed a "target dataset" in the field of sequence similarity search [40, 41]. For developing a robust algorithm, additional query dataset(s) for independent tests like the CASP sets [42] are often utilized. Rigorous development and evaluation procedures must make the composition of training, testing, and independent sets very different or non-redundant, which may avoid overestimation caused by information leakage. Hence, accurate algorithms mostly emphasized that the homology between those query sets was low during development.

Although this general scheme has been rigorous, to our knowledge, there are still problems not deeply examined yet. (1) Will using the same reference dataset to construct the PSSMs for training, testing, and independent tests cause information leakage and overestimate the accuracy? (2) In addition to the requirement of low homology between query sets for training, testing, and independent tests, will the homology between query sets and the reference dataset influence the quality of evaluation? (3) Will sequence redundancy of the PSSM reference dataset affect accuracy? The widely used UniRef90, for instance, is highly redundant because the identity between its sequences can be up to 90%.

In this study, we discovered that the general SSP development scheme might overestimate the accuracy in some steps while making underestimation in others. This situation might have made the actual accuracy of current SSP methods questionable when predicting new, novel, or machine-unlearned proteins. Consequently, we designed a strategy for the dataset preparation and evaluation of SSP methods that helps avoid the over/underestimation of accuracy and guaranteeing the performance in the face of novel proteins. We are looking forward that this strategy can be adopted by outstanding SSP algorithms nowadays. By doing so, the updated predictors will accomplish reliably high accuracy and advance all research and applications depending on accurate prediction of protein secondary structures.

## Results

### Effects of the sequence homology between training and testing query datasets on the accuracy of secondary structure prediction

In the development and evaluation of machine-learning-based SSP methods, it has long been generally assumed that the homology between query datasets for training and test will affect the reliability of predictions. Overfitting may occur if the inter-dataset homology between query sets is high. Therefore, most SSP studies highlighted low sequence identity between the query dataset for training and that for testing or independent test. According to our preliminary tests (see **Materials and Methods**), as the homology of query proteins decreased, the degree of overfitting in SSP lowered. Nevertheless, in that experiment, both the inner- and inter-dataset sequence identities of the training and testing sets were decreased, and which dominated the results was not sure. To verify this long-established assumption, we first tested the effect of homology **between** query datasets on SSP accuracy.

The experimental design is shown in Fig 1A. We adopted the dataset layout commonly applied in SSP research nowadays. Most studies use one PSSM reference dataset, and its inner-dataset sequence identity is <90% (*i.e.*, UniRef90). In this study, we used the NrPdb90-2015 as the source of reference sequences. Besides, we set up two query sets, one for training and the other for testing, in which all proteins were taken from the 2015 PDB (Protein Data Bank [43]). Finally, there were independent test datasets TS115 [44] and CASP12 [42], which comprised protein structures that were solved after January 2016 and shared ≤30% sequence identities with proteins of the 2015 PDB. Based on this arrangement, we gradually reduced the inter-dataset sequence identity between training and testing query sets using (PSI-)CD-HIT-2D [45]. With this homology reduction, the composition of the two query sets ranged from allowing duplicates (All), no duplicates (NR100), to non-redundant at a series of identity cutoffs (NR90 to NR20). Besides, in this experiment, we made ten repeats by random sampling for each homology level to calculate the average and standard deviation of accuracy. Surprisingly, as shown in Fig 1B, the homology between the query sets for training and testing did **NOT** influence SSP accuracy. No matter what the **inter-dataset** identity was, the accuracy of training, testing, and independent tests remained at their respective levels. In this test, the **inner-dataset** identities of the training and testing datasets were both 90%. We had also tested 40% and 20% identities and observed the same phenomenon–the **inter-dataset** sequence identity exerted little influence on the accuracy of SSP (Fig 1C and 1D). In this figure, accuracy was represented by the Q3 measure. We had also computed the SOV. Despite that the SOV value was generally lower than Q, the results shown in S1 Fig reached the same conclusion.

In this report, three types of SSP accuracy are exhibited. (1) Accuracy of training obtained with the query set used to train the machine learning model. (2) Accuracy of testing obtained with another set of query sequences to verify the feasibility of the model. (3) Accuracy of independent tests obtained with independent query sets sharing very low homology with other

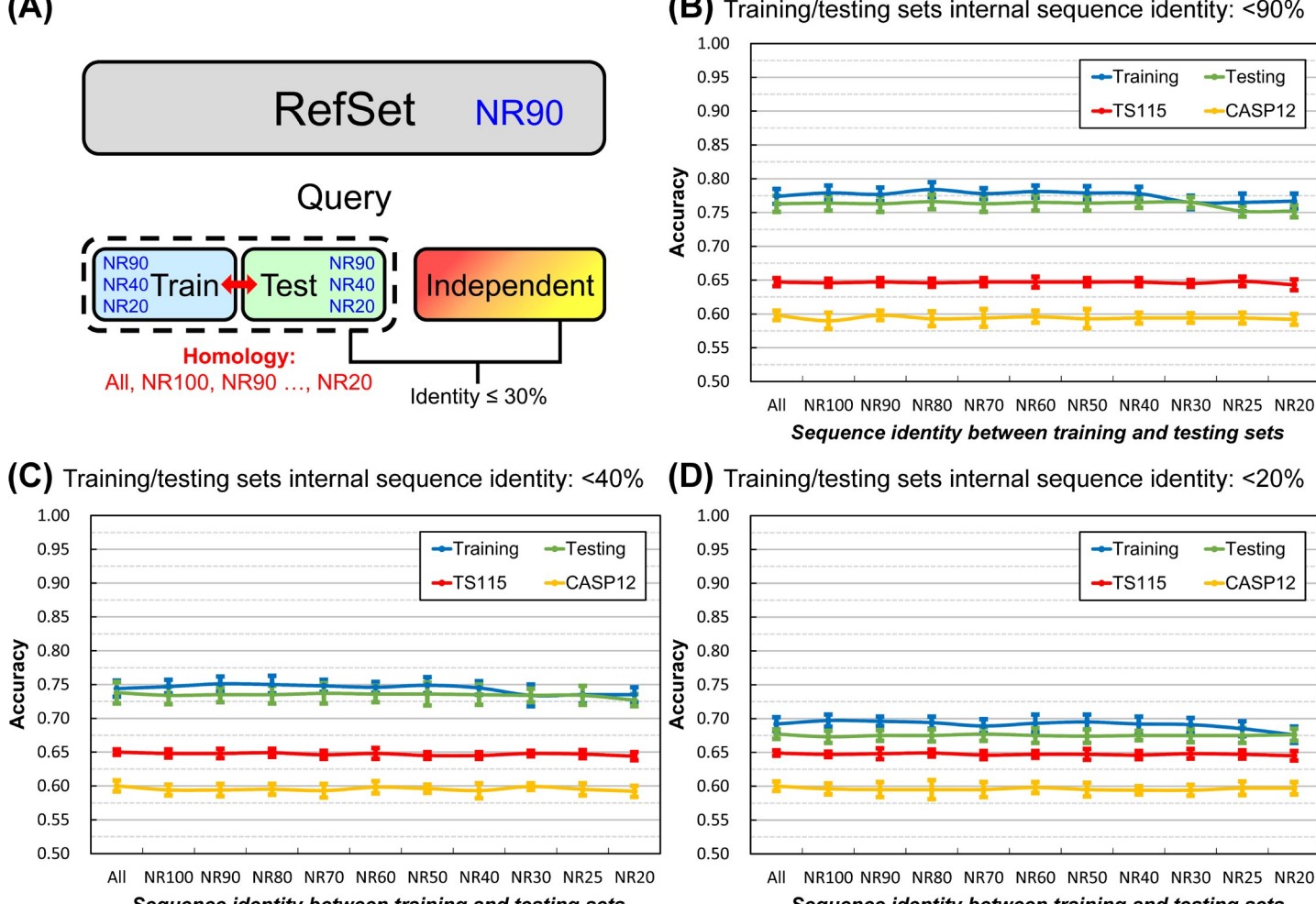

**Fig 1. Effects of the homology between training and testing query datasets on the accuracy of secondary structure prediction.** (A) The layout of datasets. The NR*x* indicates the non-redundancy of sequences in one or between two datasets, where *x* is the sequence identity cutoff. For example, the NR90 labeled on the reference dataset (RefSet) indicates that any two sequences' identity in the dataset was <90%. In this layout, the homology between training and testing query sets decreased while the individual query set's homology was fixed at 90%, 40%, or 20% identity cutoffs. All proteins in the reference and training/testing sets were obtained from the 2015 PDB. Dataset size: reference 10,000, training 250, testing 250. The independent test datasets, TS115 [44] and CASP12 [42], consisted of PDB proteins from 2016 or later with low homology with the 2015 PDB. (B) Results of decreasing **inter-dataset** homology with fixed inner dataset homology (<90%). (C) Results of decreasing **inter-dataset** homology with fixed inner dataset homology (<40%). (D) Results of decreasing **inter-dataset** homology with fixed inner dataset homology (<20%). Unlike the assumption of previous studies, the homology between training and testing query sets exhibited little influence on SSP accuracy, no matter in training, testing, or independent tests. Repeating this experiment at three inner-dataset homology levels reached the same conclusion. These independent test accuracies were lower than previous reports (**Materials and Methods**) because the reference set was much smaller than the conventional UniRef90 dataset. For reasons why the accuracy of CASP12 is lower than TS115, see **Materials and Methods** as well.

datasets used in the same test. As the final evaluation, the purpose of independent tests is to challenge the machine learning model with cases very different from those it has learned. Typically, the accuracy of training is higher than testing because the model is built from the training set. However, a significant difference between them is the sign of overfitting. Similarly, the accuracy of training/testing is usually higher than independent tests because independent tests are meant to be difficult challenges, but a big difference between them implies serious overfitting. We consider that for an SSP strategy, only the accuracy obtained with rigorous independent tests is its "true accuracy" (or the practical accuracy), while that of training or testing is just the "apparent accuracy" and is suspicious of overfitting. In other words, using any strategy,

# (A)

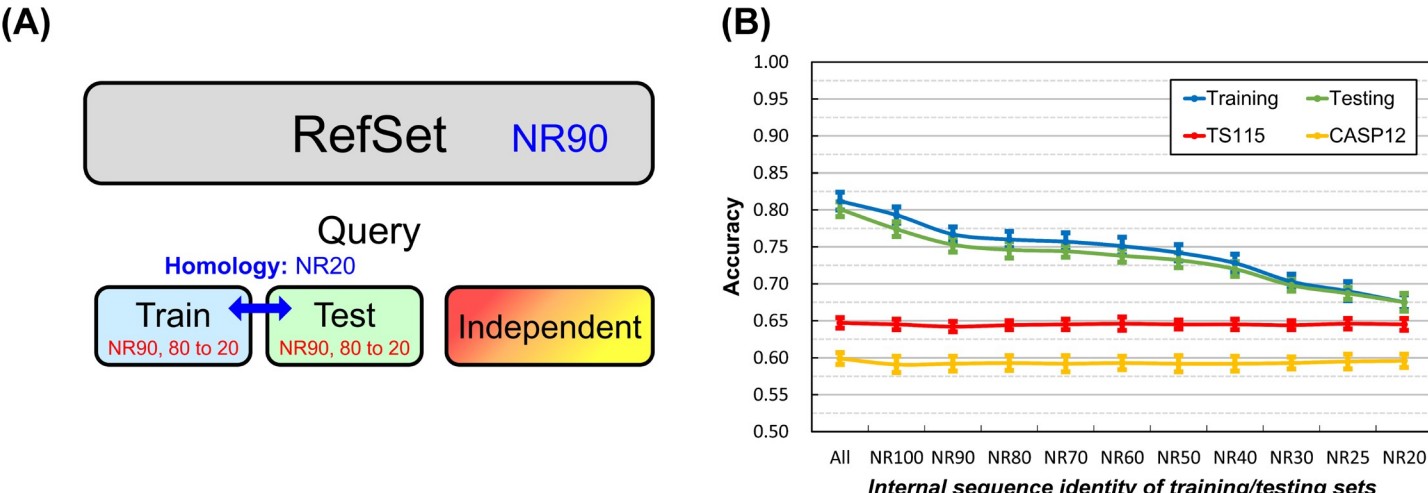

# (B)

**Fig 2. Effects of the homology within the training/testing query dataset on the accuracy of secondary structure prediction. (A)** The layout of datasets. The homology between training and testing query datasets was fixed, while the homology within each query dataset decreased. Dataset size: reference 10,000, training 250, testing 250; independent test dataset TS115: 115, CASP12: 46. **(B)** Results of fixed inter-dataset homology (<20% identity) with decreasing **inner-dataset** homology. The accuracy of independent tests remained steady as training and testing query sets' inner-dataset homology decreased; meanwhile, the accuracy of training and testing was remarkably decreased. Since the accuracy of independent test represents the practical accuracy when an SSP method faces query proteins very different from those it had learned, the fact that the accuracy of training/testing approached the accuracy of independent test indicated a reduction of overfitting. This experiment revealed that although the homology within training and testing query sets has little influence on the practical accuracy of SSP, lowering it can greatly reduce overfitting. See S2 Fig for the same trend in SOV.

only if the accuracy of independent tests is increased, the SSP is genuinely improved. Importantly, provided that the independent test accuracy is not lowered, a decrease in training or testing accuracy does not necessarily indicate a flawed strategy but maybe an improvement that suppresses overfitting.

Based on this knowledge, the results illustrated in the latter part of **Materials and Methods** and Fig 1 revealed that the dual reduction of the inner- and inter-dataset homology of the training and testing sets suppressed the overfitting of SSP but reducing just the inter-dataset homology was not the case. A reasonable explanation was that between the "inner" and "inter" dataset homology, it was the "inner" that mattered in reducing overfitting. To test this hypothesis, we applied another experimental design that fixed the homology between training and testing datasets and gradually reduced the sequence homology within each of them (Fig 2A). As expected, the overfitting of SSP was reduced as the **inner-dataset** homology of query sets decreased (Fig 2B). In addition to the Q index, we had computed the SOV score as well (see S2 Fig) and achieved the same conclusion–decreasing the **inner-dataset** sequence identity of the training/testing query sets can significantly reduce the overfitting of SSP.

## Effects of isolation of the reference proteins for training, testing, and independent test on the accuracy

This experiment aimed to answer the first question of this work: whether using the same reference dataset to generate PSSM for training, testing, and independent tests might lead to overestimation of accuracy. The experimental design is shown in Fig 3A. We still applied the typical setting of current SSP works, in which the reference set was composed of proteins sharing <90% sequence identities (source: NrPdb90-2015). The difference was that the number of reference sets ranged from one to three, according to the illustrated layouts. In each layout, we adjusted the query sets for training and testing to perform the same test at three homology

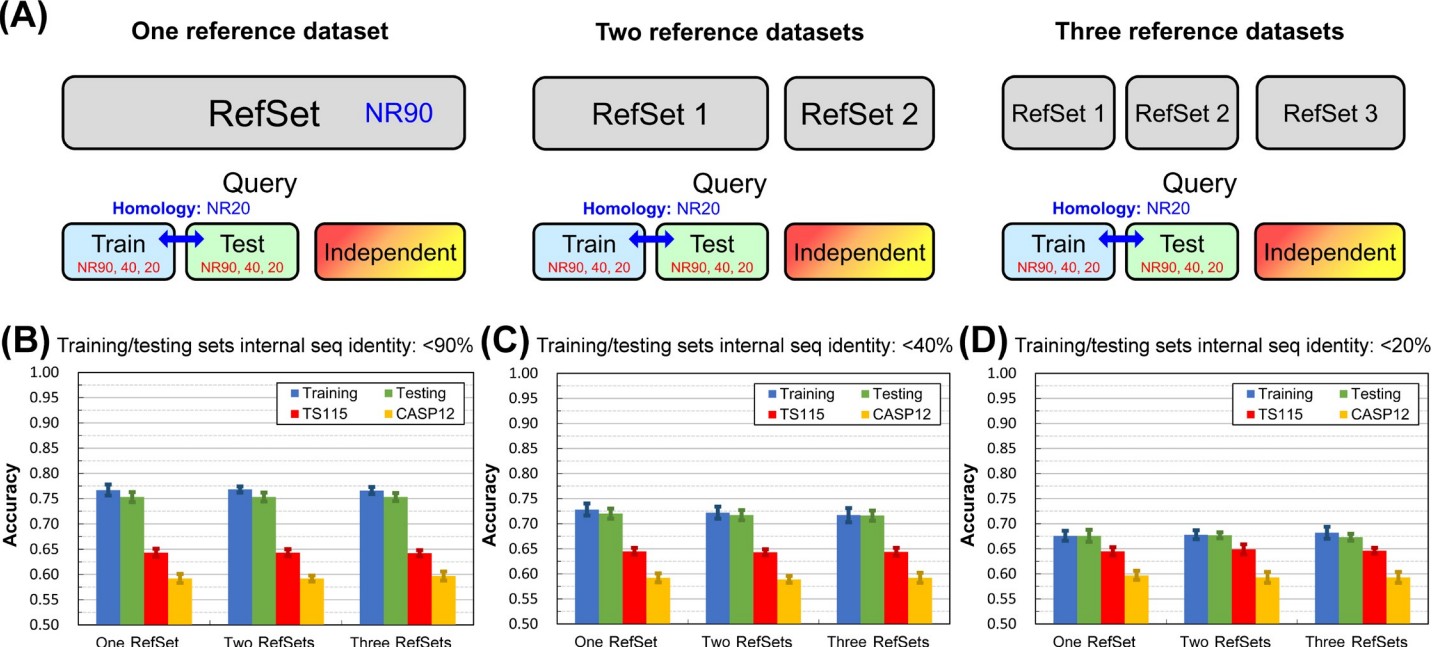

**Fig 3. Effects of the isolation of PSSM reference sequences for training and evaluations on secondary structure prediction accuracy. (A)** The layout of datasets. The PSSM reference sequences were used as a single dataset or divided into two or three different datasets. Regardless of the number, the size of each reference dataset was 10,000 sequences. The sequence identity cutoff between training and testing query sets (250 sequences for each) was fixed at 20%. Each layout was tested at three levels of inner-dataset homology of training and testing query sets. **(B)** Results obtained with training/testing sets having inner-dataset sequence identities <90%. **(C)** Results obtained with training/testing sets having inner-dataset sequence identities <40%. **(D)** Results obtained with training/testing sets having inner-dataset sequence identities <20%. The isolation of reference sequences did not reduce overfitting. Regardless of whether one or multiple reference sets were used, with a given homology cutoff for sequences within the training and testing sets, the accuracy of each training, testing, and independent test group remained unchanged (see S3 Fig for results of SOV). To conclude, using only one reference set for both training and evaluations, as most previous SSP studies did, would not cause increased overfitting.

levels, high: <90%, medium: <40%, and low: <20% sequence identities. Before this experiment, we supposed that using the same PSSM reference dataset for training and evaluations might cause information leakage such that the accuracy in training would be higher than that in evaluations, a sign of overfitting. Unexpectedly, the results shown in Fig 3 revealed that the isolation of reference sequences did not affect SSP accuracy. Hence, we concluded that using the same PSSM reference dataset for training and evaluations would not cause overfitting in SSP. However, it is noteworthy that in this experiment, overfitting was still observed, for the accuracies of training and testing were higher than those of independent tests. In this part, the conclusion of no overfitting only applied to the isolation of reference sequences. There must be other reasons for the overfitting observed in Fig 3.

## Effects of the homology between the reference dataset and the training or testing query datasets on accuracy

To figure out what other factors might influence the accuracy or overfitting, we proceeded to the second question: whether the sequence identity between the query and reference datasets would affect SSP evaluation. Nowadays, the main machine-learning feature set of state-of-the-art SSP methods is PSSM, which is generated based on alignments between the query protein and its homologs identified from the reference dataset by PSI-BLAST or HHBlits [46]. This fact implies that modern SSP methods work, at least indirectly, based on sequence homology between the query protein and the reference dataset. Therefore, we expected that the homology between training/testing query sets and the PSSM reference set would significantly affect the

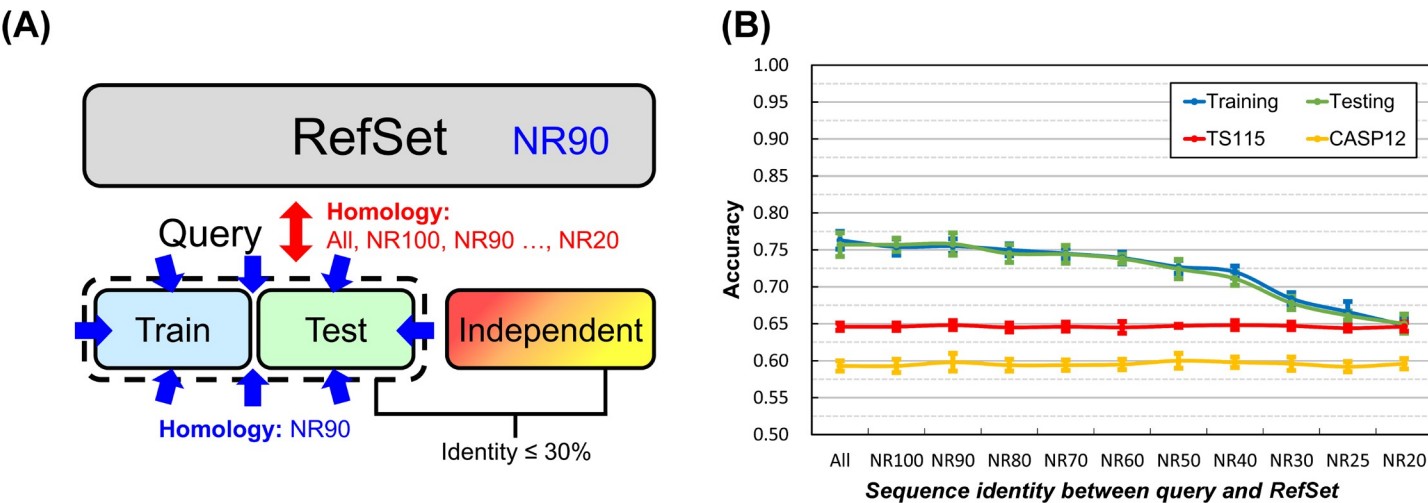

**Fig 4. Effects of the homology between query and PSSM reference datasets on the accuracy of secondary structure prediction. (A)** The layout of datasets. Only one reference dataset containing 10,000 proteins with <90% sequence identities was used. To ensure that there would be sufficient proteins to sustain the size of training and testing query sets used in this study, *i.e.*, 250 for each, the sequence identity cutoff between and within these query sets was fixed at 90%. **(B)** Accuracy of SSP obtained with training/testing query sets sharing decreasing sequence identities with the PSSM reference dataset. As expected, the homology between the query sets for training/testing and the PSSM reference dataset greatly influenced the extent of overfitting. The same influence was observed with the SOV measure (S4 Fig). To our knowledge, most SSP studies had not yet paid attention to the homology reduction between query and reference datasets, which might have led to overestimated accuracies. See **Discussion** for more information.

reliability of the evaluation of SSP methods. Specifically, the higher their homology, the severer the overfitting would be. As illustrated in Fig 4A, in this experiment, the homology between training/testing query sets and the reference dataset was gradually reduced from allowing duplicates (All), no duplicates (NR100) to non-redundant at a series of sequence identity cutoffs (NR90 to NR20). As expected, shown in Fig 4B, while the practical accuracy (accuracy of independent tests) remained steady, training and testing accuracy declined rapidly as the sequence homology between query and reference datasets decreased. In addition to revealing the importance of homology reduction between the training/testing query set and the PSSM reference dataset in developing SSP methods, these results also explained why the accuracy of independent tests performed with the TS115 and CASP12 independent datasets had remained almost unchanged in all experimental conditions applied by now. The homology between these independent query sets and the source dataset of the reference sequences, the Pdb-2015, was very low (see **Materials and Methods**). If the query-reference sequence homology is indeed a critical factor in determining the accuracy of PSSM-based SSP methods, then the low homology between TS115/CASP12 and the PSSM reference datasets used in our experiments would naturally result in the stably low accuracies.

## Effects of the sequence homology within the PSSM reference dataset on accuracy

According to our computational protein science experiences, the sequence redundancy of experimental datasets usually influences the quality of results and the reliability of developed algorithms a lot. However, most modern SSP algorithms were developed and evaluated using UniRef90, a highly redundant dataset, for generating PSSM. We wondered whether the PSSM reference dataset's sequence redundancy would interfere with the precise evaluation of SSP algorithms. The experimental design is shown in Fig 5A, where the homology of the reference set was reduced from allowing duplicates to allowing only sequences sharing <30% identities.

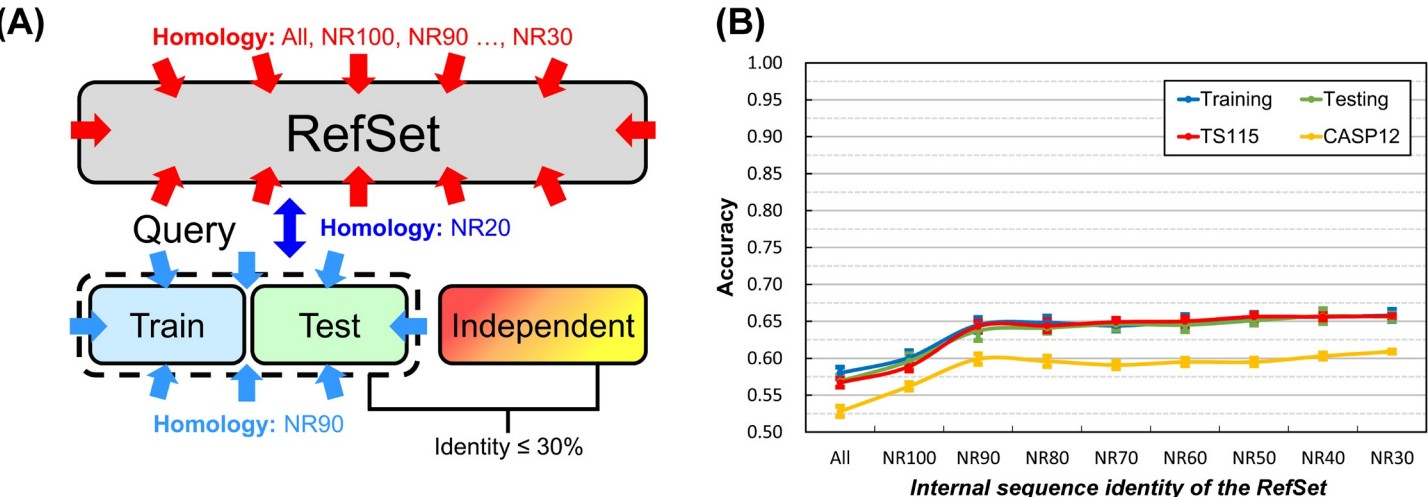

**Fig 5. Effects of the homology reduction of the PSSM reference dataset on the accuracy of secondary structure prediction. (A)** The layout of datasets. The homology of the PSSM reference dataset was reduced with a series of sequence identity cutoffs; the lowest was 30% because when 25% or 20% were applied, the remaining sequences would be insufficient to sustain the required dataset sizes (Fig 1A). The homology between training/testing query sets and the reference dataset was manipulated to be <20% sequence identities. The inter- and inner-dataset identity cutoff of training and testing query datasets were both 90%, set high for preserving sufficient sequences. **(B)** The SSP accuracy obtained at different homology levels of the reference dataset. The overfitting of prediction in training and testing was much suppressed because of the fixed low query-reference dataset homology. More importantly, the accuracy **increased** as the homology of reference sequences lowered. The same conclusion applied to SOV (see S5 Fig). This phenomenon had not been clearly reported before our study. See Fig 6 for advanced tests.

We expected that when the reference dataset's homology was high, the degree of overfitting would be high. Although it turned out challenging to determine whether the overfitting was affected by the homology of the reference set, it was surprisingly discovered that the homology reduction of the reference set **increased** SSP accuracy in either training, testing, or independent tests. As shown in Fig 5B, all these accuracies rose as the homology of the reference set decreased. Since this phenomenon had not been reported and (1) the utilized reference dataset was tiny when compared with UniRef90, and (2) the predictor used here was our in-house implementation, we repeated this test using UniRef90 as the source of reference sequences and state-of-the-art SSP algorithms as the predictor (see the next subsection).

### The accuracy of state-of-the-art SSP methods tested with PSSM reference datasets of decreasing sequence homology

Several highly accurate SSP methods were utilized in this large-scale experiment, including four 3-state and three 8-state algorithms. Because the independent test datasets applied in this study were composed of structures solved after Jan. 2016, the PSSM reference dataset used here was the UniRef-2015, a collection of proteins sequenced by Dec. 2015. The dataset layout of Fig 5A was still applied, except that the training and testing procedures were not applicable because the prediction model and program of these state-of-the-art SSP methods had all been pre-trained and compiled (source code for training not available). The UniRef100-2015 and UniRef90-2015 non-redundant sets, where 100 and 90 indicate their identity cutoffs, were obtained from the UniRef server, and they comprised 70.5 and 38.2 million sequences, respectively. Homology reduction was performed on the UniRef90-2015 to generate reference datasets with identity cutoffs lower than 90%. The lowest sequence identity reference dataset was the UniRef30-2015 because when applying an identity cutoff lower than 30%, the remaining sequences were insufficient to meet the large-scale requirement of this experiment. The size of the reference datasets was fixed at one million sequences. Similar to the result of our in-house

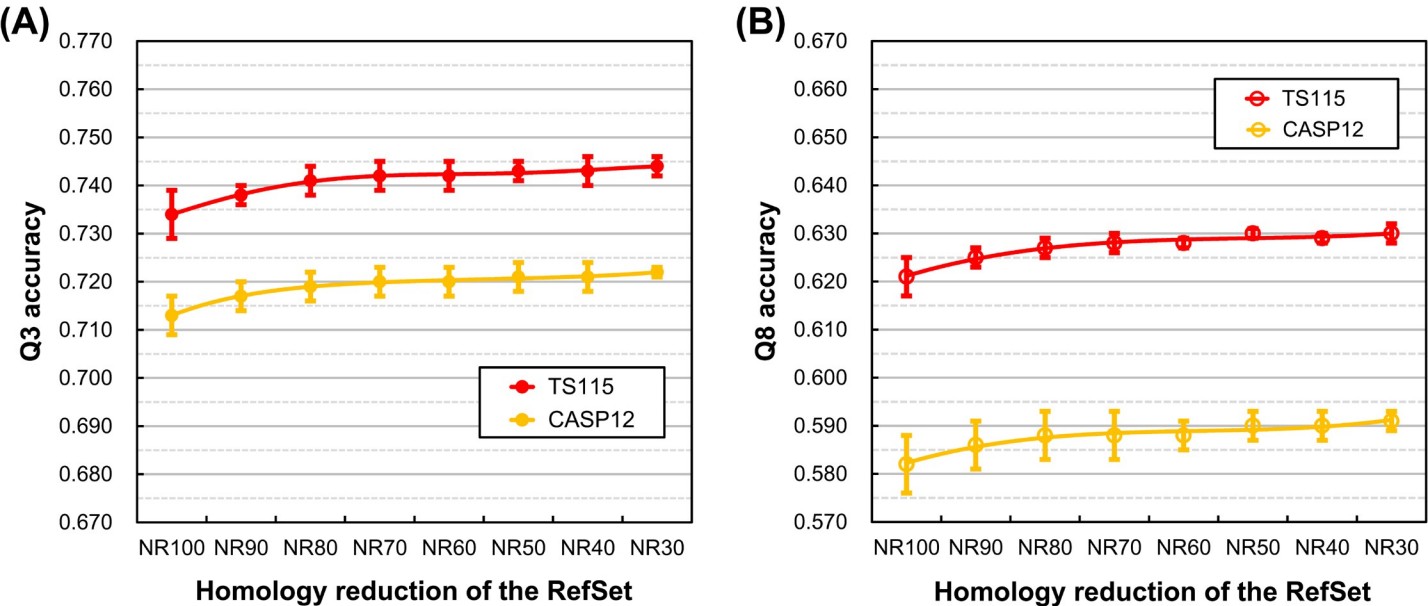

**Fig 6. Accuracy of seven state-of-the-art secondary structure prediction methods evaluated using homology reduced PSSM reference datasets. (A)** The average three-state SSP accuracy of state-of-the-art methods. **(B)** The average eight-state SSP accuracy of state-of-the-art methods. Advanced SSP methods including three-state algorithms PSIPRED [27], Scorpion [32], Spider2 [33] and SpineX [31], as well as eight-state DeepCNF [35], RaptorX [30], and SSpro8 [29] were utilized. The three-state test (A) was performed using all methods, and the eight-state test (B) was performed using eight-state algorithms. The query sets were independent test datasets TS115 and CASP12. The source of reference sequences was the UniRef of 2015. In addition to homology reduction, the reference dataset's size was fixed at one million sequences by random sampling in each test group. With random sampling, for each SSP method, the experiment was repeated five times to obtain the average and standard deviation of its accuracy at all homology levels. Finally, the statistics of the tested methods were averaged. The results reveal that the homology reduction of PSSM reference sequences can **raise** the practical accuracy of SSP methods.

predictor evaluated with small PDB-based datasets, the independent test accuracy of these well-developed SSP methods evaluated with large UniRef reference datasets also increased as the homology of reference dataset was reduced, no matter quantified with the residue-based Q3 and Q8 (Fig 6) or the segment-based SOV3 and SOV8 measures (S6 Fig).

## The proposed development and evaluation strategy for SSP methods

We have discovered that reducing the internal homology of training/testing query datasets and the query-reference inter-dataset homology can suppress SSP overfitting. Besides, reducing the homology of reference sequences helps improve the practical accuracy. Although the inter-dataset homology between training and testing query sets showed no significant influence on SSP, it might still be appropriate to keep it low because the overfitting seemed slightly decreased at low homology (see Fig 1 and **Discussion**). Based on these discoveries, here we propose a standard strategy for the development and evaluation of SSP methods, that is, the sequence homology between and within all experimental datasets should be rigorously reduced, the homology between independent test dataset and PSSM reference dataset included, as illustrated in Fig 7A. In practice, current homology reduction algorithms cannot adequately support or are not recommended for sequence identity cutoffs below 50–30% [45, 47, 48]; therefore, the operating cutoff we recommend is 30%.

To test the proposed strategy, we performed a large-scale experiment. The source of reference sequences was UniRef-2015, the training and testing query sets were sampled from Pdb-2015, and the independent test datasets were TS115, TS416, and the CASPs. The sequence identity between and within the developmental datasets (training, testing, independent test,

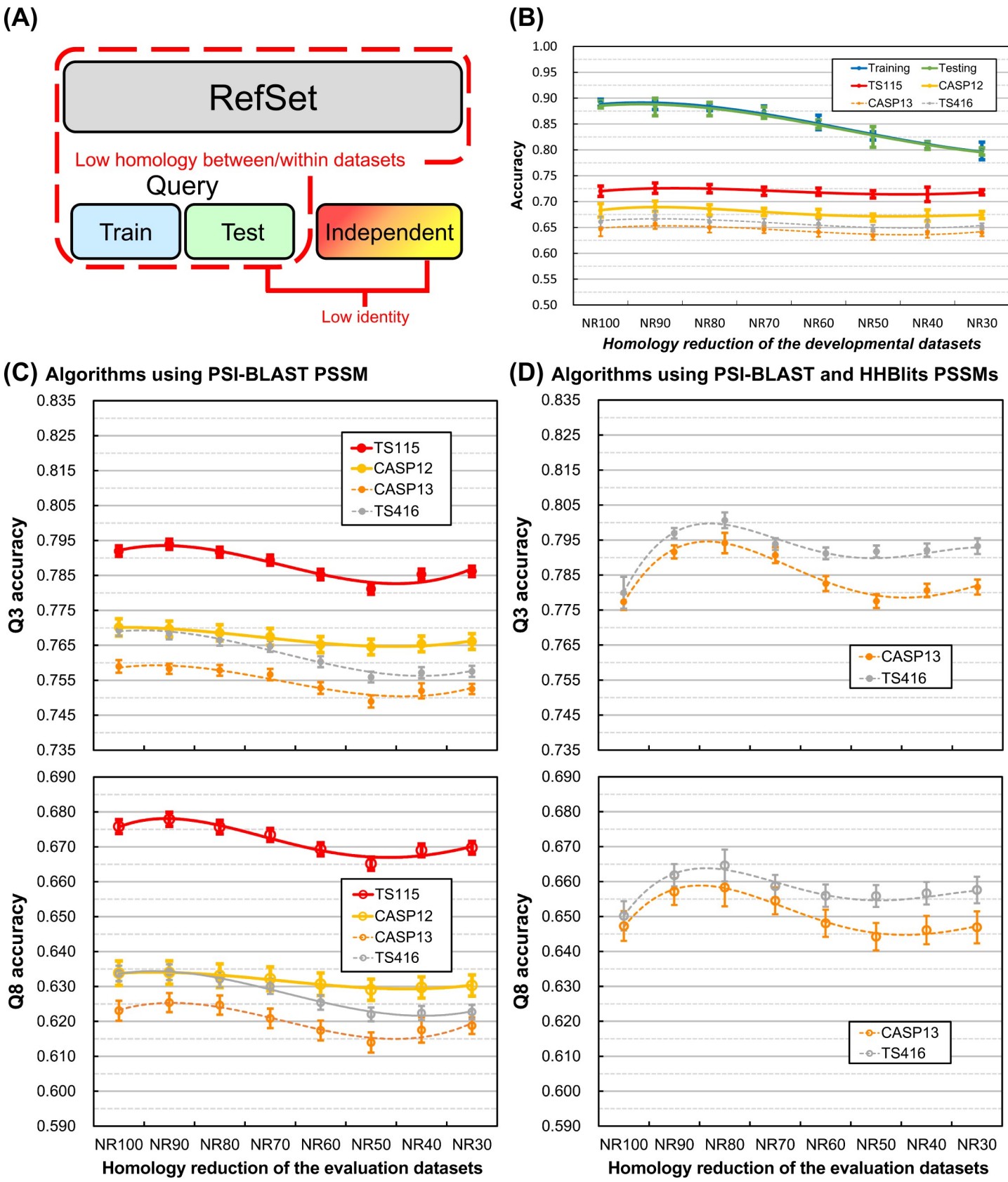

**Fig 7. The proposed development and evaluation strategy for SSP methods and its feasibility tests. (A)** The layout of datasets. We propose the homology between and within all developmental datasets, meaning the training and testing query sets and the PSSM reference dataset, should be as low as possible. Besides, the homology between the independent test dataset and the developmental datasets should also be low. A 30% sequence identity cutoff for the homology reduction of datasets is recommended. **(B)** Feasibility test of the proposed strategy performed with decreasing homology of developmental datasets. Although the 30% identity cutoff is recommended, we still performed this test with a series of cutoffs. It is clear that at the 30% identity cutoff, our in-house SSP implementation had the lowest degree of overfitting. Interestingly, the accuracy of independent tests went up and then dropped. **(C)** The accuracy of state-of-the-art SSP methods (PSSM generator: PSI-BLAST) assessed with query and reference datasets of decreasing homology. All the tested methods were trained with proteins deposited in PDB before 2016, and the applied independent test datasets were all composed of proteins deposited after Jan. 2016. Every identity level was tested ten times. The Q3 was averaged over all tested methods, including DeepCNF, Psipred3, RaptorX, Scorpion, Spider2, SpineX, and SSpro8. The Q8 was averaged over eight-state methods DeepCNF, RaptorX, and SSpro8. **(D)** The accuracy of state-of-the-art SSP methods (PSSM generators: PSI-BLAST and HHBlits) assessed with query and reference datasets of decreasing homology. Algorithms tested in (D) were MUFOLD-SS, NetSurfP-2, Porter 5, and Spider3 (3 repeats). Since they were released after 2017, the TS115 and CASP12 were not applicable as independent test sets. The TS416 and CASP13 independent datasets, which comprised proteins deposited in PDB mostly after 2018 and were very different from Pdb-2018 (<25% sequence identities), were applied. Regardless of whether the SSP algorithms utilized only one or both of the PSI-BLAST and HHBlits PSSMs to make predictions, in general, the observed accuracy went to the highest point when the sequence identity of evaluation datasets fell between 80% and 90% and then dropped with fluctuations.

and reference datasets) was gradually reduced from 100% to the recommended 30%. The results plotted in Fig 7B (see S7 Fig for SOV) indicated that at the recommended 30% homology level, the degree of overfitting of the established prediction model was the lowest among all tested levels because the gap between the accuracy of training/testing and the accuracy of independent tests was the smallest. It is noteworthy that the accuracy of independent tests rose and dropped as the identity of developmental datasets decreased. Among the key factors: (1) the sequence identity within/between query datasets, (2) the query-reference dataset identity, and (3) the internal identity of the reference dataset, because (1) might exert little influence on the practical accuracy of SSP (Figs 1 and 2) and the decrease of (3) was capable of increasing the accuracy (Fig 5), we conjectured that it was the decrease of (2) that counteracted the accuracy-improving effect of the decrease of (3). Before examining this conjecture in detail (see **Discussion**), we first verified whether this phenomenon was universal. After repeating this experiment using state-of-the-art SSP methods published either before 2015 [27, 29–33, 35] (PSSM generator: PSI-BLAST) or after 2016 [34, 36–38] (PSSM generators: PSI-BLAST and HHBlits), similar fluctuations were observed, no matter the accuracy was measured in three- or eight-state (Figs 7C, 7D and S7). The algorithms published in different years or using different types of PSSM exhibited slightly different behaviors, but the general trend was that assessing SSP methods with developmental datasets sharing 80–90% sequence identities would produce higher accuracies than with datasets sharing ≤50% identities. These results imply that developing or evaluating an SSP algorithm without strict homology reduction between the query and PSSM reference datasets will likely lead to accuracy overestimation.

## Discussion

### The influence of homology of experimental datasets on the accuracy and evaluation of secondary structure prediction

**Sequence homology between the query datasets for training and testing.** Perhaps different from most SSP researchers' expectations, Fig 1 shows that the homology between training and testing query sets has little effect on SSP accuracy. The idea of the necessity of low sequence identity between training and evaluation query sets applied in most SSP studies may not be entirely correct. However, it must be noted that before the formal experiments, we had selected the ANN, which exhibited low overfitting in SSP, as the machine learning algorithm (**Materials and Methods**). If the selected algorithm had a high degree of overfitting, the homology between the training and testing sets might still affect the performance of SSP. For example, we also tested the decision tree and SVM using the same procedure of Fig 1. As shown in Fig 8, their training accuracy was higher than testing, which was then significantly

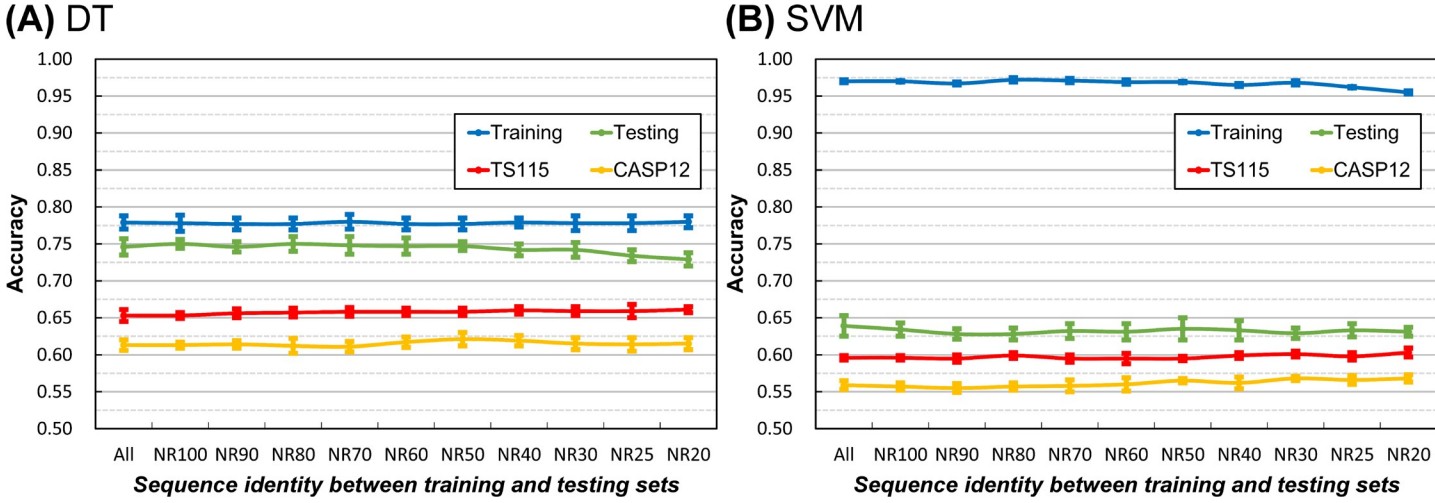

**Fig 8. Effects of the homology between training and testing query datasets on SSP accuracy obtained with decision tree or support vector machine.** (A) Results of the decision tree. (B) Results of the support vector machine. This test's procedure was the same as the experiment of Fig 1, but the inner-dataset sequence identity cutoff of the training and testing query sets was fixed at 40%, and the machine learning algorithms were the decision tree and SVM. Even when an algorithm prone to overfitting was applied, the inter-dataset homology between the training and testing query sets had little influence on SSP accuracy, consistent with ANN results. Although the influence was minor, the homology reduction between these query sets still contributed to the decrease of overfitting. At very low sequence identity, either the gap between the accuracy of training and testing reduced or the accuracy of independent tests increased slightly.

higher than the accuracy of independent tests, indicating severe overfitting. Nevertheless, as the homology between the two query sets decreased, the gap between training/testing and independent test accuracies became slightly smaller. Similar trends were observed with SOV measures (S8 Fig). Therefore, it is still recommended that the homology between the training and testing sets should be low.

**Sequence homology within the query dataset.** We discovered that the **internal** homology of the training and testing query sets played a major role in decreasing the overfitting of SSP (Fig 2). Interestingly, the declining trends of overfitting in training and testing were the same. In that experiment, the inter-dataset sequence identity between training and testing query sets was <20%. The two query sets were so different, but the overfitting associated with them decreased in the same trend as their internal homologies were reduced independently, implying that some common factor influenced the overfitting. We speculated that the most likely factor was the homology between the query sets and the PSSM reference dataset. As we kept reducing the internal homology of training and testing sets, more and more sequences sharing high identities with other sequences were also eliminated, perhaps leading to a side effect, that is, the homology between the sequences remaining in these query sets and the reference dataset was "passively" reduced.

**Sequence homology between the query and reference datasets.** Fig 4 revealed that reducing the homology between query and reference datasets remarkably decreased overfitting. The query-reference dataset homology has not been explored before our study, but it is now demonstrated very important, for it greatly affects the reliability of the evaluation of an SSP method. To further test the importance of this homology, we did an interesting experiment. The homology between the query sets was fixed at a 20% sequence identity cutoff. On purpose, the identity cutoff between the training query set and the reference dataset was manipulated to be 20%, while that between the testing query set and the reference dataset was very high: 90%. As a result, the accuracy of testing was much higher than that of training (see S9 Fig). Future SSP researches should attach great importance to this query-reference dataset homology.

**Sequence homology within the reference dataset.** It had been noticed that using Uni-Ref50 as the PSSM reference set would make SSP more accurate than UniRef90 [30], but the impact of homology reduction of the reference dataset on SSP has not been thoroughly examined. According to our findings (Figs 5 and 6), some previous SSP methods' accuracy might be underestimated because of the inadequate homology reduction of the PSSM reference dataset. However, since this underestimation is far weaker than the overestimation effect of the query-reference homology (Fig 4), it is more likely that the accuracy of a previous method was overestimated if the independent test was not delicately designed. See the next subsection for detailed discussions about the reference dataset.

## On the role of the reference dataset in SSP and the accuracy improvement which it may bring about

**Isolation of the reference sequences and the information leakage.** The reference dataset's fundamental role in the current general SSP methodology is to provide homologous sequences for producing the PSSM. To our knowledge, all SSP algorithms developed after PSIPRED used only one reference dataset during development and evaluations. As we raised the three major problems of SSP at first (see **Introduction**), we speculated that using the same reference dataset for development (training and testing) and evaluation (independent test) might cause information leakage and overestimate the accuracy. However, we observed at last that no matter the reference sequences were used as a single set or divided into different sets for training, testing, and independent tests, the accuracy was not affected. Our explanation for this unexpected result refers to the nature of the PSSM.

The PSSM is a substitution matrix in logarithm scores [49] for each residue of the query sequence [28]. For a query protein, the PSSM is generated based on the sequence alignments between the query and its homologs identified using a sequence similarity search algorithm, *e. g.*, PSI-BLAST and HHBlits. In other words, the PSSM encodes the overall homology between the query and reference sequences. A hidden message behind this methodology is that the accuracy of a query protein is correlated with its homology with reference sequences. Therefore, no matter in development or evaluation, as long as the overall homology between a given query set and the reference dataset(s) was the same, the accuracy would not be affected by whether the reference sequences were isolated. Hence, using one reference dataset both to develop and evaluate will not overestimate SSP accuracy because of information leakage. To further examine this putative conclusion, we repeated the experiment of Fig 3 by changing the machine learning algorithm from ANN to decision tree or SVM (S10 Fig). Again, the accuracy was not affected by whether the reference sequences were isolated.

**Effects of the size of PSSM reference dataset on SSP accuracy.** Recently we have discovered that the size of the PSSM reference set has a sigmoid relationship with SSP accuracy [50]. In that study, which only used UniRef datasets, we observed that the Shannon information entropy of the PSSM generated by a small reference set was lower than that generated by a large one, and, between reference sets of different sizes, the PSSM entropy was highly correlated with the accuracy. The average entropy of PSSMs generated with the TS115 and a reference set containing 9,327 or 1.2 million UniRef90 proteins was 0.60 or 2.47, respectively, and the Q3 obtained with the 1.2-million-protein reference set was 8.1% higher than the small set (see the Fig 7 of [50]). In the present work, to delicately control the homology of datasets, we used the non-redundant dataset prepared by PDB in most experiments, which greatly limited the size of reference datasets. The average entropy of the PSSMs generated with TS115 based on a reference set of 10,000 NrPdb90-2015 proteins was 0.52 (the result of Fig 5), and that based on a 1.0-million-protein UniRef90-2015 reference set was 2.34 (the result of Fig 6). The

Q3 of TS115 obtained with the 1.0-million-protein reference set was 9.5% higher than the small set. Consistent with [50], the information entropy and accuracy increased as the reference dataset expanded.

In addition to providing supporting data to our earlier work, we find it **NOT** true that using a larger reference dataset will lead to a more reliable prediction. If homology reduction of the developmental datasets is not performed carefully, a larger reference set will cause severer overfitting (see the last part of **Materials and Methods**), meaning the constructed predictor performs well for proteins similar to the learned ones but performs poorly for the unlearned. There is no need to use a massive reference set to improve accuracy, especially when many SSP methods' accuracy has been shown to reach saturation if the reference dataset exceeds 5 million proteins [50].

**Effects of homology reduction of the PSSM reference dataset on SSP accuracy.** Using large datasets, recently we have also found that, for identity cutoffs ≤90%, homology reduction of the PSSM reference dataset can **improve** SSP accuracy [50]. In the present study, when the cutoff was expanded to 100% or even canceled, the same conclusion was reached (Figs 5 and 6). Regarding why the homology reduction of the PSSM reference set could improve SSP, we have hypothesized that the reference set's homology level would affect the quality of the PSSM, which ultimately influenced SSP accuracy [50]. Previously, we quantified the quality of PSSM by information entropy and found it highly correlated with SSP accuracy. Now we have a supposition based on several facts. First, a diverse set of sequences would produce a PSSM with lower information entropy than a conserved set [50]. Second, a higher information entropy means more information is encoded in the probability distribution of a variable, and the PSSM is not only a probability distribution but also the main machine-learning feature set in SSP. Third, the size of the hit list PSI-BLAST uses to generate a PSSM is limited (default: 500 hits). Our supposition is that the diversity of the limited amount of homologs retrieved by PSI-BLAST would determine the quality/entropy of the PSSM, which would then affect the performance of machine learning and the SSP accuracy. The first support of this supposition is the high correlation between SSP accuracy and the entropy of PSSMs generated either by PDB or UniRef reference datasets (Table 1). Furthermore, given a fixed size of the hit list, the diversity of the homologs retrieved from a small reference dataset should be higher than that from a large dataset. If so, as the homology of the reference set reduces, it can be expected that the correlation between PSSM entropy and SSP accuracy calculated from a small reference set will be higher than that calculated from a large set. Indeed, as shown in Table 1, the correlation coefficient between these two variables computed from 10,000-protein homology-reduced reference

**Table 1. Correlation between the information entropy of PSSM and SSP accuracy as the sequence homology of the reference dataset decreases.**

| Source and size of the reference dataset | Measure | Homology level of the reference dataset | | | | | | | | | Corr[d]. |
|---|---|---|---|---|---|---|---|---|---|---|---|
| | | **All** | **NR100** | **NR90** | **NR80** | **NR70** | **NR60** | **NR50** | **NR40** | **NR30** | |
| **Pdb-2015, 10K[a]** | Entropy | 0.326 | 0.398 | 0.515 | 0.513 | 0.531 | 0.525 | 0.573 | 0.564 | 0.582 | 0.987 |
| | Q3 | 0.567 | 0.589 | 0.643 | 0.644 | 0.649 | 0.650 | 0.656 | 0.656 | 0.657 | |
| **UnrRef-2015, 1M[a]** | Entropy | N/A[c] | 2.264 | 2.340 | 2.382 | 2.426 | 2.453 | 2.489 | 2.492 | 2.510 | 0.968 |
| | Q3 | N/A[c] | 0.734 | 0.738 | 0.741 | 0.742 | 0.742 | 0.743 | 0.743 | 0.744 | |
| **UnrRef-2015, 5M[b]** | Entropy | N/A[c] | N/A[b] | 2.623 | 2.660 | 2.708 | 2.739 | 2.788 | 2.811 | 2.845 | 0.949 |
| | Q3 | N/A[c] | N/A[b] | 0.800 | 0.803 | 0.803 | 0.804 | 0.805 | 0.805 | 0.806 | |

[a]Results of the TS115 independent test of Fig 5 (10,000 proteins) and Fig 6 (1 million proteins).

[b]Results of the TS115 independent test of the Fig 6 of [50] (5 million proteins). The NR100 identity level was not tested.

[c]Not available because the UniRef database only provided non-redundant sets.

[d]Pearson's correlation coefficient.

sets was higher than those computed from 1-million-protein and 5-million-protein homology-reduced reference sets.

## The contradicting effects of reducing the query-reference dataset homology and the internal homology of the PSSM reference dataset

In most SSP works, the homology reduction between query and reference datasets and the homology reduction of the reference dataset were not performed. Now we find that the former will decrease the apparent accuracy (Fig 4), whereas the latter can improve both the apparent and practical accuracy of SSP (Figs 5 and 6). In Fig 7, a rise-and-drop trend of accuracy was observed. We conjectured that the contradicting effects of these two homology reductions caused this particular trend. Importantly, this trend implied that the accuracy of some SSP methods might have been overestimated.

Firstly, the main difference between Figs 6A, 6B and 7C experiments is the homology level between the independent test query sets and the PSSM reference dataset. In Fig 6A, the query-reference homology was not manipulated, and the accuracy increased along with the homology reduction of the PSSM reference dataset. In Fig 7C, the query-reference homology and the homology within the reference dataset decreased together, and the accuracy fluctuated. These results revealed that the homology between the query set (independent test set included) and the PSSM reference dataset greatly influences the reliability of the evaluation of SSP methods.

Secondly, based on the results of Fig 7C, when the sequence identity between and within the evaluation datasets is 80–90%, the observed SSP accuracy will reach the highest point, while the lowest point will occur at sequence identities ≤50%. Most SSP methods were developed and evaluated based on the UniRef90 PSSM reference dataset, and the query-reference dataset homology was not manipulated, meaning that the sequence identity between and within some developmental datasets was high. For example, using CD-HIT-2D to eliminate homologs of TS115 and CASP12 from the UniRef90-2015 with a 40% identity cutoff would remove 131 thousand sequences. The high homology of datasets, especially between independent test query set(s) and the PSSM reference dataset, might cause overestimated SSP accuracy.

Finally, to verify whether the homology between the independent test query set and the reference dataset will truly affect the reliability of SSP evaluation, we took the UniRef-2017 and UniRef-2019 datasets, which had increasingly more homologs of TS115 and CASP12 than the UniRef-2015 did, to repeat the experiment of Fig 7C and 7D. As shown in Fig 9, the accuracies of TS115 and CASP12 tests at a 90% query-reference sequence identity cutoff raised year by year. It was because the amount of closely-related homologs of the independent test sets in the PSSM reference dataset increased over time. Differently, at a 30% cutoff, the accuracy remained stable because most homologs of the independent test sets had been eliminated from the reference dataset. Since the tested SSP methods were all trained with proteins far-related from the TS115 and CASP12, this result also revealed that even if the homology between the query sets for training and the independent test is ultimately low, an overestimate of the accuracy will still occur if the homology between the independent test query set and the PSSM reference dataset was not adequately reduced.

## Details of the proposed strategy for future SSP methods

The core concept of the proposed strategy for developing and evaluating an SSP method is to make all datasets highly non-redundant in sequence homology, whether within or in between, including the query sets for training, testing, and independent tests, as well as the reference dataset for generating PSSMs (Fig 7A). In particular, the homology between the independent query set and all other datasets must be low. The recommended homology cutoff within and

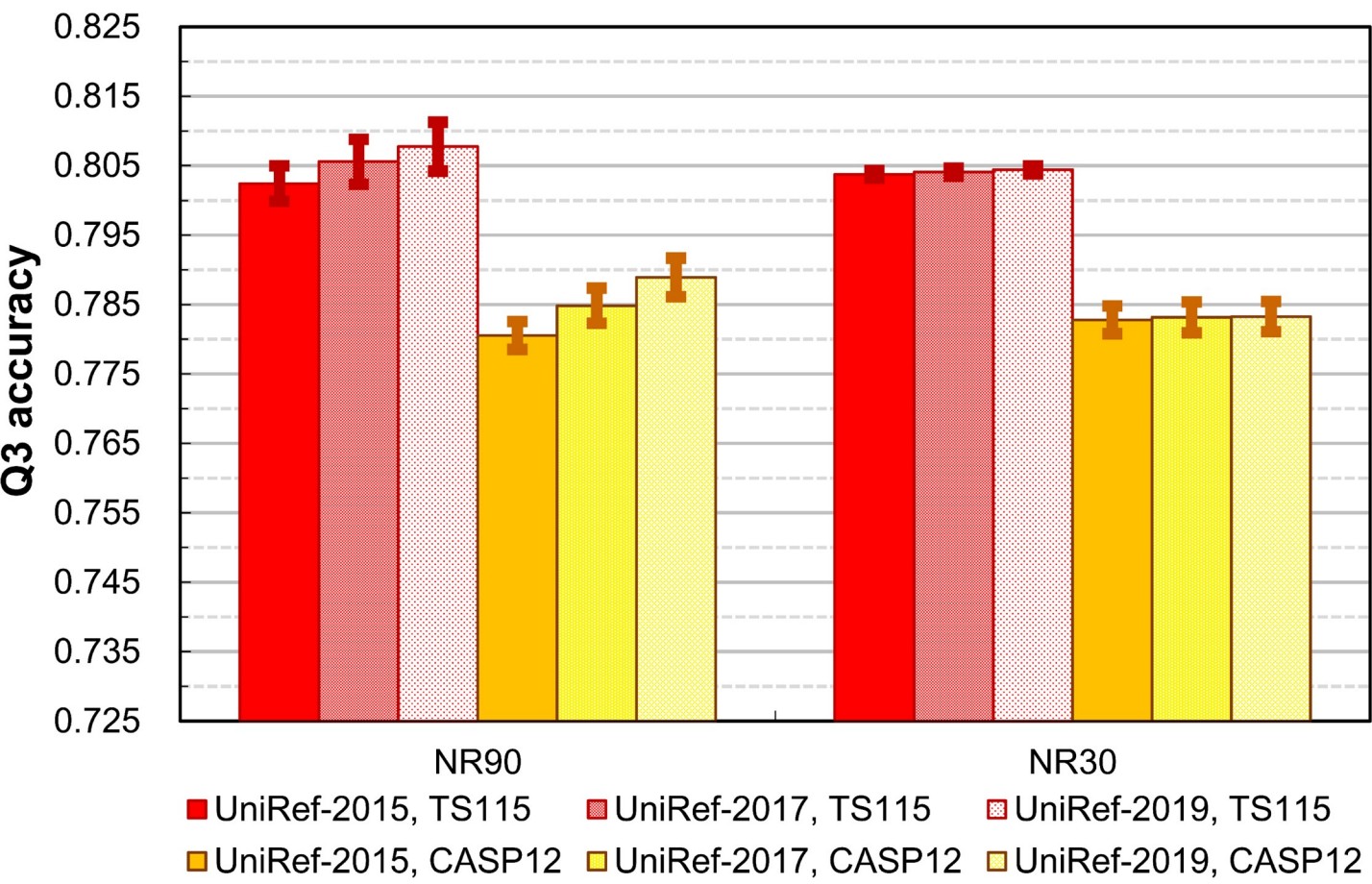

**Fig 9. Advanced feasibility tests for the proposed development and evaluation strategy for SSP methods.** In these two homology reduction test groups, the layout of Fig 7A and the seven advanced SSP methods developed before 2016 were applied (4 repeats). The PSSM reference dataset used in each test was fixed by random sampling to be 5 million proteins, the SSP accuracy saturating dataset size reported in [50]. The TS115 and CASP12 were composed of novel protein structures determined after Jan. 2016; hence, they had few homologs in either the UniRef90-2015 or UniRef30-2015. As time went by, they had more and more homologs deposited in the UniRef, and their overall homology with UniRef-2017 and UniRef-2019 gradually increased. When the reference dataset was adequately homology-reduced against the query sets, *e.g.*, the NR30 group, the accuracy was stable no matter the UniRef of which year was used. Contrarily, if the homology reduction were insufficient, the observed accuracy was not reliable, even if the query set was advertised as an "independent dataset".

between datasets is 30% sequence identity. It is not necessary to use a massive reference set for generating PSSMs, especially when the low homology of datasets is not established. A reference dataset of 5 million sequences is sufficient to saturate an SSP method's accuracy and can significantly cut down the time cost (refer to [50]). During the evaluation, using the same PSSM reference dataset used to train the predictor will not make the accuracy overrated. Multiply repeating an experiment by random sampling is urged, for it helps prevent arbitrary conclusions based on just one particular combination of sequences. In addition, to avoid overweighting small proteins or underweighting large ones when calculating the average accuracy over a set of query proteins, the length-weighted average and the residue-based micro average (**Materials and Methods**) are promising alternative algorithms to the conventional arithmetic mean.

## Materials and methods

### Experimental datasets

The original protein sequence and structural data were obtained from the PDB [51] and UniRef [39]. Well-prepared independent test sets from the third parties were also applied.

According to the experiments' requirements, two or more sequence datasets might be prepared to have a certain non-redundancy level of homology from each other. In such a case, our primary strategy was to sample subsets from a non-redundant dataset of the required homology level. The sequence identity non-redundancy of the source dataset would define the inter-dataset sequence identity of the subsets sampled from it. For instance, because any two proteins in an $x$% identity non-redundant source dataset shared $<x$% sequence identity, if A and B were different subsets sampled from it, then any two sequences from A and B would have $<x$% identity. Another way was to utilize inter-dataset homology reduction software like the CD-HIT-2D [45], beneficial when the datasets had different origins (PDB *versus* UniRef) or when the required inter- and inner-dataset homology levels were different.

**Homology reduced subsets of the Protein Data Bank.** The non-redundant datasets with sequence identity cutoffs from 100% to 30% officially released in 2015 by the PDB were obtained from [43]. These official PDB non-redundant sets were prepared by heuristic BLAST-based clustering. To ensure the homology of experimental datasets, three homology reduction methods, including the USEARCH, CD-HIT, and MMseqs2 [45, 47, 48], were iteratively applied to these non-redundant sets until no more sequences could be removed by any method at the given sequence identity cutoffs. According to the sequence identity cutoff, the produced datasets were named the NrPdb100-2015 (70,789 proteins), NrPdb90-2015 (34,533 proteins) . . ., and NrPdb30-2015 (15,059 proteins). Since the lowest identity non-reduction set officially prepared by PDB was only 30%, the same three-method iterative homology reduction procedure was performed on the NrPdb30-2015 dataset to make the NrPdb25-2015 (11,449) and NrPdb20-2015 (5,092 proteins). The full PDB entity dataset released in 2015 is abbreviated to Pdb-2015 (273,920 proteins) in this report.

**Homology reduced subsets of the UniRef.** The non-redundant UniRef datasets with identity cutoffs 100% and 90% officially released in Dec. 2015, namely the UniRef100-2015 and UniRef90-2015, were obtained from the UniProt [39]. They contained 70.5 and 38.2 million proteins, respectively. UniRef datasets with lower homology were required to fulfill our experimental design, and those with identity cutoffs ranging from 80% to 30%, termed the UniRef80-2015, UniRef70-2015 . . ., and UniRef30-2015 (7.0 million proteins), were obtained from [50].

**Inter-dataset homology reduction.** It is very convenient to use inter-dataset homology reduction software to remove redundant sequences between two datasets. However, attention should be paid to whether the redundant sequences are adequately removed. For instance, after using the (PSI-)CD-HIT-2D [45] to perform inter-dataset homology reduction to remove from dataset B the proteins sharing identity $\geq$40% with any protein in dataset A, a subset of B (denoted as B') will be produced. One may expect that the sequence identity between A and B' is $<$40%. If it is the case, when A is in turn subjected to inter-dataset homology reduction using the same program to remove proteins sharing sequence identity $\geq$40% with any protein from B', none should be eliminated. However, we found that this expectation was not always accomplished even when the sequence identity was calculated as local identity, at least when the (PSI-)CD-HIT-2D was applied. For ensuring that the redundant homologous sequences between two datasets are removed, we would like to suggest a "two-way" inter-dataset homology reduction procedure, that is, after performing B–A = B', the A–B' = A' should also be done such that the produced A' and B' will have the required inter-dataset non-redundancy. To apply this procedure, the CD-HIT-2D is adequate for sequence identity cutoffs $\geq$40%, while the PSI-CD-HIT-2D supports cutoffs $<$40%.

**Independent test datasets.** In this study, we have implemented a machine-learning SSP predictor. In most experiments, the reference dataset and query sets for training and testing the predictor were proteins from the Pdb-2015. To perform stringent independent tests, we

utilized independent datasets composed of novel proteins released after Jan. 1st, 2016, including the TS115 (115 proteins) established by [44], and the CASP12 (46 proteins) and CASP13 (43 proteins) obtained from the biannual meeting of Critical Assessment of Structure Prediction techniques [42]. The TS115 was claimed to have ≤30% sequence identities with proteins deposited in the PDB before 2016 [44]. We used PSI-BLAST to compute homology and found that TS115 and the CASPs all had a ~25% median sequence identity with Pdb-2015. However, 61% of TS115 proteins had identities ≥20% with Pdb-2015, and this proportion in CASPs was only ~50%. Hence, the homology between CASPs and Pdb-2015 was lower than that between TS115 and Pdb-2015. The CASP12/13 mainly covered novel proteins deposited in the PDB between 2015 and 2018. Following the data preparation procedure of TS115 [44], we established another independent test dataset, TS416 (416 sequences; S1 File), comprising PDB 2019 and 2020 proteins that shared <25% sequence identity from proteins deposited in PDB by Dec. 2018. The inner-dataset sequence identity of the TS416 was also <25% (homology reduced by USEARCH, PSI-CD-HIT, and MMseqs2).

## Random sampling and multiple repeats of experiments

To be rigorous and to ensure the stability of results, most experiments were repeated several times using random sampling techniques. The detailed random sampling procedure varied from experiment to experiment, but the basic flow was the same. For easy explanation, here we take the case of Fig 1 as an example, in which we needed a non-redundant set of sequence identity <90% as the reference dataset (10,000 proteins), a query set for training (250 proteins), and a query set for testing (250 proteins); besides, the two query datasets must share sequence homology from high to low. The procedure is described as follows,

1. Randomly select 10,000 proteins without replacement from the NrPdb90-2015 as the reference dataset.

2. Let $x$ stand for the homology level between the query sets, and the initial $x$ is set to "All" (see below for meaning).

3. Randomly select 250 proteins from NrPdb$x$-2015 without replacement as the query set for training and another 250 proteins as the query set for testing.

4. Use the query set for training to build a predictor and the one for testing to test the predictor. The reference dataset used in this process is the one prepared in Step 1.

5. Independent tests. Use the independent test datasets as query sets to evaluate the predictor.

6. Repeat Steps 1–5 for $n$ times to obtain $n$ versions of accuracy data; each version comprises the Q3 and SOV3 of the training, testing, and independent tests.

7. Repeat Steps 1–6, and change the $x$ to 100, 90, . . . to 20 (%) in turn to obtain the accuracy data as the homology between training and testing query sets decreases.

Through these steps, we obtain $n$ versions of accuracy data at each homology level. At any given level, for each accuracy measure, the mean and sample standard deviation can be calculated. The number of repeats ($n$) for all experiments was 20 unless otherwise specified. In this procedure, the NrPdbAll-2015 is exactly the Pdb-2015, which contains many redundant sequences. The NrPdb100-2015 is a non-redundant set in which every sequence, perhaps sharing high identities with others, is unique. As for the other NrPdb$x$-2015 datasets, $x$ represents the homology cutoff in percentage sequence identity.

## Applied secondary structure prediction methods

**Implementation of the modern SSP methodology.** An SSP predictor was created for experiments by implementing the methodology of current SSP algorithms as described previously [27, 35], that is, utilizing PSSM as the main feature set and amino acid information as additional features, setting a window for encoding interactions of nearby residues into the final feature set for every residue, and using machine learning to construct the prediction model. The PSSM we utilized in most experiments was the conventional PSI-BLAST PSSM and 8-state SSE codes. In some specific experiments, the HHBlits PSSM was also applied to make 8-state SSE predictions. The window used in this study was five residues. Previously as we studied a protein rearrangement phenomenon known as the circular permutation [52, 53] and examined what combination of machine learning and predictive features performed best in predicting suitable bioengineering sites for proteins to be circularly-permuted, we had developed an integrated machine learning system. This system, abbreviated as iMLS, supported the artificial neural network (ANN), decision tree (DT), and support vector machine (SVM), and these algorithms could be applied in a conventional way or in combination with bootstrapping to form random forest(s) [54]. In the present study, the iMLS was utilized to create our in-house SSP predictor.

**Utilized state-of-the-art secondary structure prediction methods.** Although the predictor we implemented was based on the general SSP methodology, our machine learning system was for common use but not optimized for SSP. To test whether our experimental conclusions applied to various SSP algorithms, we also utilized highly accurate SSP algorithms published **before 2016**. The emphasis on "before 2016" is because the independent test datasets we used were composed of novel protein structures released after Jan. 2016. Developed before 2016, these algorithms' prediction models were not trained with proteins similar to the independent datasets. Although they were very suitable for this study, we could only obtain their compiled programs with pre-trained prediction models. Without source codes, it was impossible to retrain models. Therefore, these algorithms could not be applied in all experiments and were only utilized in the final large-scale ones. In alphabetical order, the applied advanced SSP algorithms are the DeepCNF (v1.02) [35], PSIPRED (v3.3) [27], RaptorX (v1.0) [30], Scorpion (v1.0) [32], Spider2 [33], SpineX (v2.0) [31], and SSpro8 (v5.2) [29]. Several modern algorithms published after 2016 were also tested, including the MUFOLD-SS (v2.0) [36], Net-SurfP-2 [37], Porter 5 [38], and Spider3 [34]. Among these recent works, NetSurfP-2 used only HHBlits, while the other three used both the PSSMs generated by PSI-BLAST and HHBlits as predictive features.

## Usage of PSI-BLAST and HHBlits, the PSSM generation engines

All SSP algorithms applied in this work utilized PSI-BLAST to perform sequence similarity searches for building PSSM. Because they recruited different versions of PSI-BLAST with different settings, which may influence the quality of experiments, we had to unify the version (*i.e.*, NCBI PSI-BLAST v2.6.0) and parameter settings. We referred to the common settings of the applied SSP algorithms and PSI-BLAST default settings to set unified parameters (see S1 Table for the original and unified settings). To verify that the algorithms were applied correctly and that the unified PSI-BLAST settings did not degrade their performance, we have preliminarily measured their accuracy. As shown in S2 Table, the accuracies calculated using the unified settings were close to those reported in previous studies when these algorithms were evaluated with equivalent datasets [35, 44]. Some of the applied state-of-the-art SSP algorithms also used HHBlits to generate PSSM. The version of HHBlits we used was v3.3.0.

## Computation of secondary structure prediction accuracy

In addition to the widely used Q accuracy, we also calculated SSP accuracy using the SOV (segment overlap) measure [55, 56]. The conventional Q accuracy is defined as the number of correctly predicted residues divided by the number of predicted residues. The SOV is not based on residues but on secondary structure segments to compute SSP accuracy for a given protein. Compared with the Q accuracy, the SOV measure is considered a more rigorous SSP assessment method because it can evaluate the overall quality of SSP for a protein and reduce the noise produced by individual residues [55–57]. SSP accuracies can be computed based on the three- or eight-state secondary structure classifications. Because the experiments of this study were very time-consuming, we only performed the three-state SSP. Nevertheless, according to our previous study [50], for each SSP algorithm that supports both three- and eight-state SSPs, the trend of three-state accuracy (Q3/SOV3) and eight-state accuracy (Q8/SOV8) was the same under various conditions. The difference was that eight-state accuracies were lower than three-state ones. The focus of the present report is the quality of the development and evaluation procedure for SSP algorithms rather than the difference among accuracy measures. In order to reduce the time cost and prevent distracting the readers, the quality of eight-state SSP was only assessed in the preliminary verifications (*e.g.*, S2 Table) and some final large-scale experiments.

## Statistical analyses

**Significance analysis with the *p*-value.**　In this work, experiments were repeated several times by random sampling. When it was necessary to check whether the difference of a measure observed between two groups was statistically significant, the *p*-value was calculated. First, the Shapiro-Wilk test was used to check the normality of the accuracy values of each group. Next, if the normal distribution was verified, an *F*-test was performed to determine the equality of the variances of the two groups. If the groups were verified to come from populations with equal variance, the Student's *t*-test was performed to calculate the *p*-value, or the Welch's *t*-test was used instead.

**The weighted average and micro-average of accuracy measures.**　When performing SSP accuracy evaluation, typically, multiple query sequences are used. Hence, the final accuracy value is an average. In most SSP researches, the average accuracy is calculated using the standard arithmetic mean, that is, the summation of accuracy from all proteins divided by the number of proteins. The size of query sequences may be very different, but the arithmetic mean treats long and short sequences equivalently. Perhaps an ideal way to calculate the accuracy for a set of query sequences is the weighted average, where the weight for each sequence is its length. Alternatively, instead of using protein as the unit, we may use residue as the unit to calculate the average accuracy for a set of proteins. In this study, SOV values were calculated using the weighted average $(\overline{SOV})$, while the Q was calculated using residue as the unit $(\overline{Q_{res}})$. The formulas are shown below,

$$\overline{SOV} = \frac{\sum_{q=1}^{n} (len_q \times SOV_q)}{\sum_{q=1}^{n} len_q} \qquad (1)$$

$$\overline{Q_{res}} = \frac{\sum_{q=1}^{n} Nres_a(q)}{\sum_{q=1}^{n} len_q} \qquad (2)$$

where $q$ represents a query protein, $n$ denotes the number of query proteins, $len_q$ and $SOV_q$ stand for the length and SOV of $q$, respectively, $Nres_a(q)$ is the number of residues of $q$ that are

accurately predicted. The $\overline{Q_{res}}$ weighs equally all residues and prevents underweighting large proteins or overweighting smaller proteins. The protein-based Q average and residue-based $\overline{Q_{res}}$ are analogous to the macro-average accuracy and micro-average accuracy commonly used to assess machine-learning-based predictors. The weighted average and micro-average are both capable of reducing the problem caused by the length imbalance of query proteins and may better reflect the actual accuracy of SSP than the conventional arithmetic mean.

**Information entropy of the PSSM.** The Shannon information entropy, also known as the disorder or uncertainty of a set of data, measures the amount of information in a variable [58]. In the case of a multi-value or multi-state variable, the Shannon entropy (*S*) is given by,

$$S = -\sum p_s \log_2(p_s) \tag{3}$$

where *s* denotes a specific state or value of the variable, and $p_s$ is the observed probability of *s* in the probability distribution of the variable. A traditional PSSM for a residue position in a query protein contains 20 values, each meaning the observed probability (based on the query-reference alignments) of this position to be occupied by a specific amino acid. The PSI-BLAST generates a PSSM in two formats, one in bit score and the other in probability. Since PSSM is the probability distribution of a 20-state variable by nature, we supposed that the Shannon entropy could quantify the information abundance of a PSSM, and we had applied this idea in our previous study on the speed improvement in SSP [50]. In the present work, the Pearson's correlation coefficient between the entropy of PSSMs and the accuracy of SSP was computed based on their respective weighted average values, where the weight for each protein was its sequence length (see Table 1).

## Determination of the most applicable machine learning for this study

The iMLS system supported several machine learning algorithms [52, 53]. To choose suitable ones for implementing SSP, we challenged several machine learning algorithms with training/testing query datasets sharing decreasing homology. This preliminary test's layout is illustrated in Fig 10, which involved a non-redundant PSSM reference dataset, a query dataset for training the prediction model, a query dataset for testing the model, and query datasets for performing independent tests. The source dataset for the PSSM reference sequences was NrPdb90-2015. The training and testing query sets (source: Pdb-2015) were prepared to have inner- and inter-dataset identities both smaller than the given cutoffs. The query sets for independent tests were TS115, CASP12/13, and TS416. This experiment was repeated ten times by random sampling. In each repeat, some proteins were randomly selected from the NrPdb90-2015 to be the PSSM reference set, and some others randomly from the NrPdb*x*-2015 dataset (*x* stands for the sequence identity cutoff) to be the training/testing query sets. The sizes of the reference dataset, the query set for training, and the query set for testing were 10,000, 250, and 250 proteins, respectively.

Fig 11 showed that no matter for ANN, DT, or SVM, SSP accuracies on the independent test datasets were similar and steady (see S11 Fig SOV data). The accuracy of TS115, CASP12, TS416, and CASP13 was approximately 65%, 60%, 58%, and 57%, respectively. These values were lower than those of the TS115 and CASP12 tests listed in S2 Table because of the much smaller reference set size used here (10,000 *versus* 3.82 million proteins). The homology between CASPs/TS416 and Pdb-2015 was lower than between TS115 and Pdb-2015; meanwhile, the accuracy on CASPs/TS416 was lower than that on TS115, implying the influence of datasets' homology on SSP. Among these algorithms, ANN showed the smallest extent of overfitting because its training and testing performances were very similar (Fig 11A). Contrarily, overfitting was a severe problem for the SVM (Fig 11E). Combined with the bootstrapping

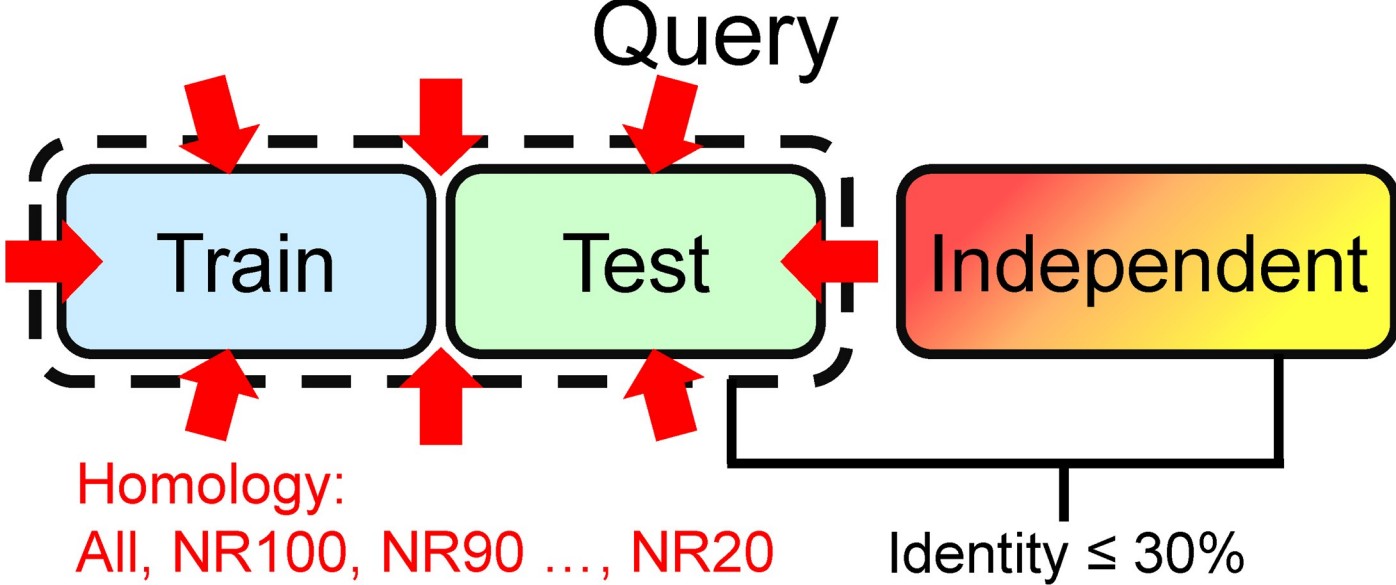

**Fig 10. Experimental design for measuring the SSP accuracy of machine learning algorithms as the sequence homology of training and testing datasets decreases.** In this layout, the PSSM reference set was a 90% sequence identity non-redundant dataset. The sequence homology of the source data of query sets was decreasing such that any sequences either in the same query set or between the training and testing query sets shared homology lower than the given identity cutoffs. The "All" means that the homology reduction is not applied, or otherwise, the number after "NR" indicates the identity cutoff of the homology reduction. Proteins in the reference dataset and training/testing query sets were obtained from the Pdb-2015. The query sets for independent tests were TS115, CASP12/13, and TS416.

technique, the overfitting of these machine learning algorithms could be significantly reduced (compare Fig 11A, 11C, 11E and Fig 11B, 11D, 11F). For all algorithms, the accuracy of testing decreased as the identity of query sequences lowered, and the accuracy approached the level of independent tests when the identity cutoff was lower than 30%. We supposed that the decrease in the accuracy of testing was not an indicator of declining performances but was, in reality, an improvement because the approaching between the accuracies of testing and independent tests stood for the reduction of overfitting. On the premise of low overfitting, ANN was the most accurate among the three algorithms. We also tested whether combining these algorithms would increase performance. As demonstrated in Fig 12, such combination exerted minimal improving effects on the accuracy of independent tests (see S12 Fig for similar results of SOV). Besides, the extent of overfitting was not reduced but just averaged after the combination. For instance, combining the low-overfitting ANN and high-overfitting SVM, the overfitting of the final prediction model lies between the two algorithms. According to these results, in this study, we decided to use ANN with bootstrapping to form a random forest of 60 ANN "trees" as the final machine learning scheme to implement the modern SSP methodology. The eight-state SSP accuracy of this combination was also assessed, and its trends in training,

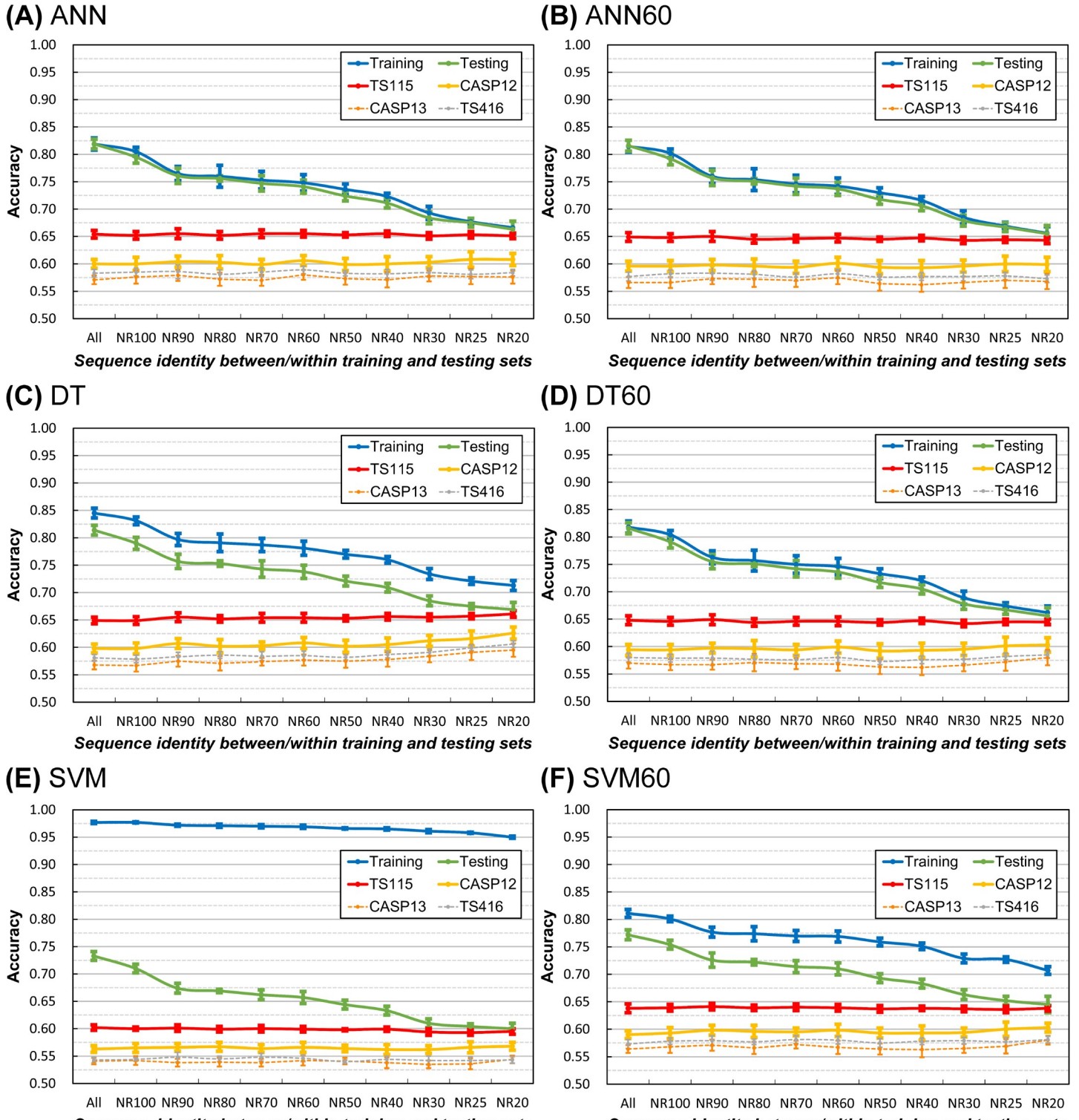

**Fig 11. The SSP accuracy of machine learning algorithms obtained with training and testing datasets sharing decreasing levels of sequence homology.** (**A**) Artificial neural network. (**B**) Random forest of 60 artificial neural networks. (**C**) Decision tree. (**D**) Random forest of 60 decision trees. (**E**) Support vector machine. (**F**) Random forest of 60 support vector machines. By applying bootstrapping, random forests of different machine learning models were established. The data and feature sets were first randomly sampled into subsets to train small models, which then made the final prediction by vote. Bootstrapping is typically applied with decision trees to form the traditional random forest. The machine learning system utilized in this study [52, 53] can make random forests of algorithms other than the decision tree. The layout of this experiment is shown in Fig 10. At each identity cutoff, the Q3 of training, testing, and independent tests were measured (10 repeats). Among these machine learning algorithms, ANN exhibited the lowest overfitting. SOV was also computed (S11 Fig), and its trends were similar to the trends of Q3 shown here.

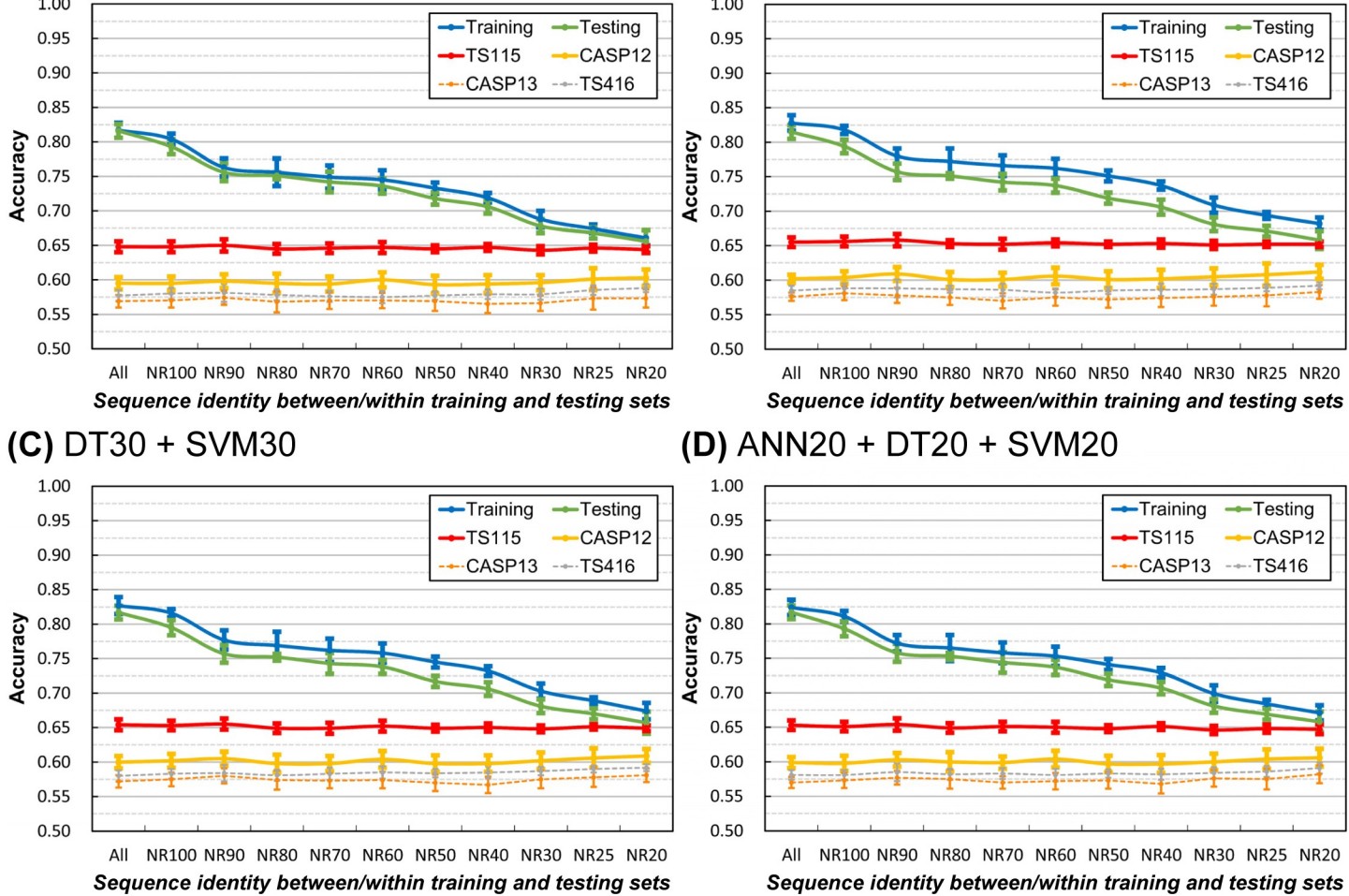

**Fig 12. The SSP accuracy of combined machine learning algorithms obtained with training and testing datasets sharing decreasing levels of sequence homology.** (**A**) Random forest of 30 artificial neural networks and 30 decision trees. (**B**) Random forest of 30 artificial neural networks and 30 support vector machines. (**C**) Random forest of 30 decision trees and 30 support vector machines. (**D**) Random forest of 20 artificial neural networks, 20 decision trees, and 20 support vector machines. The layout of this experiment was the same as in Fig 10. Using the iMLS system's bootstrapping function, random forests of several different machine learning algorithms were constructed. For fair assessments, the number of "trees" in every forest was set to be 60. Although combining different algorithms helped average out the extent of overfitting (compare the gaps between the blue and green curves shown in Fig 11 and this figure), the improvement in accuracies evaluated by independent tests was minor.

testing, and independent tests were very similar to those of the three-state accuracy but only lower (see S13 Fig). The accuracies on CASP12, CASP13, and TS416 datasets were very close, and their standard deviation bars in the graph overlapping one another. For clarity, the results of CAPS13 and TS416 were omitted in this report except for the final large-scale tests.

## Determination of the type of PSSM for experiments

The HHBlits [46] has been increasingly applied in recent SSP works to generate the PSSM homology profile [34, 36–38]. Using the selected machine-learning framework (a random forest of 60 bootstrapping ANNs), we tested the suitability of HHBlits-PSSM for this study. The genuine homology profile generated by HHBlits is encoded as a hidden Markov model (HMM). The formula for transforming the HMM score ($h$) of a given residue position (saved

in the _hhm.ffdata file by default) into a PSSM score ($p$) is,

$$p = 2^{-h/1000} \qquad (4)$$

Alternatively, the PSSM of a sequence can be obtained from the.hhr output file of HHBlits by combining the multiple sequence alignment parser program of Porter 5 (process-psi.sh) [38] and the traditional substitution scoring matrix algorithm [28, 49]. The results diagramed in S14 Fig indicated that, based on reference datasets of 10,000 proteins, using HHBlits-PSSM could improve prediction accuracy by approximately 2.5% when compared with using the PSSM generated by PSI-BLAST (refer to Fig 11B); meanwhile, the trends of the accuracy curves of training, testing, and independent tests remained similar to those of PSI-BLAST PSSM. Before using either PSI-BLAST or HHBlits for homology searches, the reference dataset must be formatted into a specialized format. The dataset-formatting procedure of PSI-BLAST is straightforward and rapid, which, using one CPU, typically took less than 0.5 sec for a dataset of 10,000 proteins (disk space: 3.2MB). However, because of the complicated multiple alignment analyses conducted by the HHBlits procedures, formatting a 10,000-protein dataset by HHBlits using its default settings cost 1.3 days on average when 12 Intel Xeon 3.33GHz

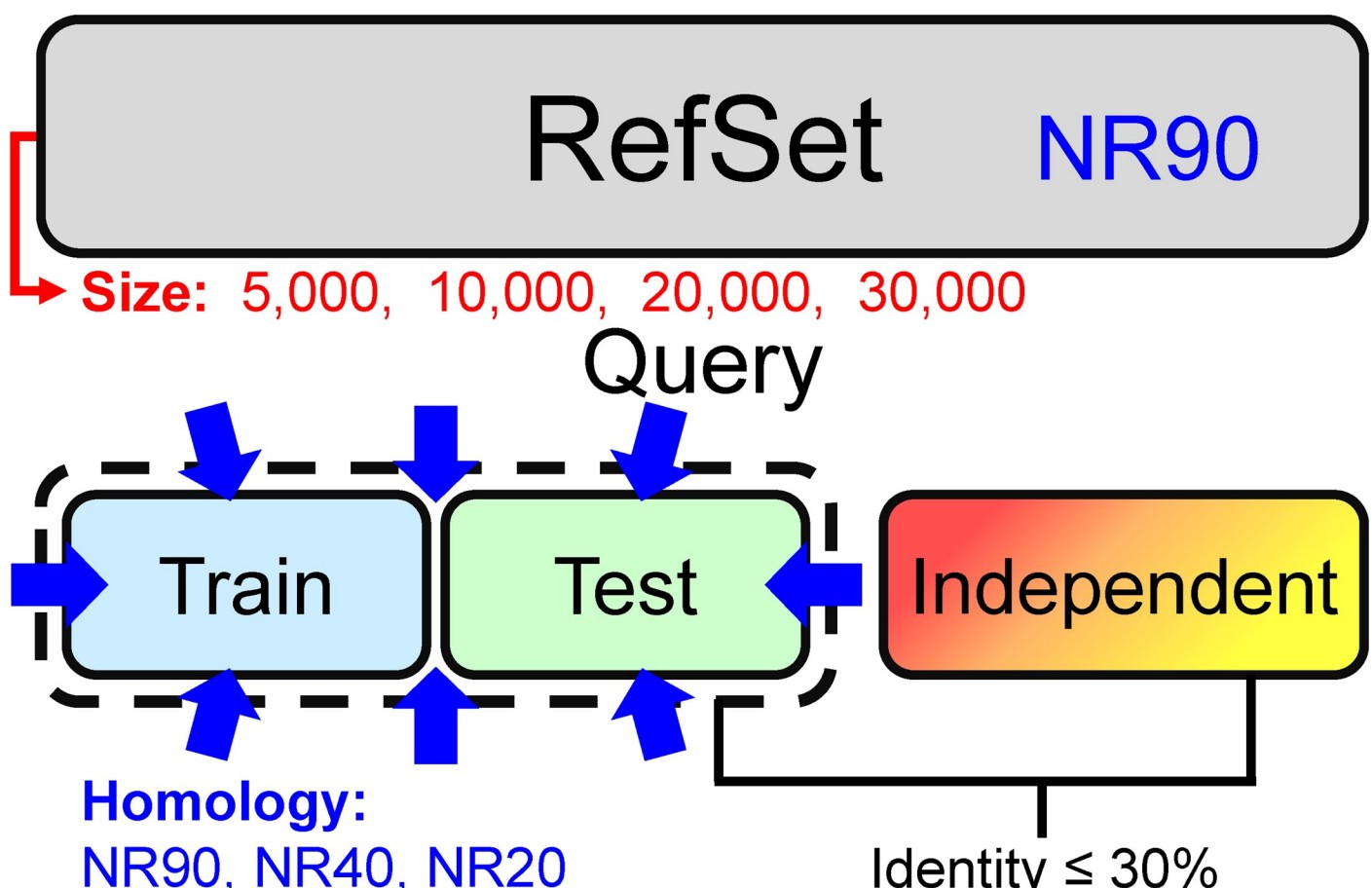

**Fig 13. Experimental design for measuring SSP accuracy under different reference dataset sizes.** The focus of this experiment was to determine the influence of the reference dataset size on SSP accuracy. The maximum size was 30,000 proteins because the source of the reference sequences, *i.e.*, NrPdb90-2015, contained only 34,533 proteins. The sequence identity of training and testing query sequences was also adjusted to see whether the reference dataset size's influence would be affected.

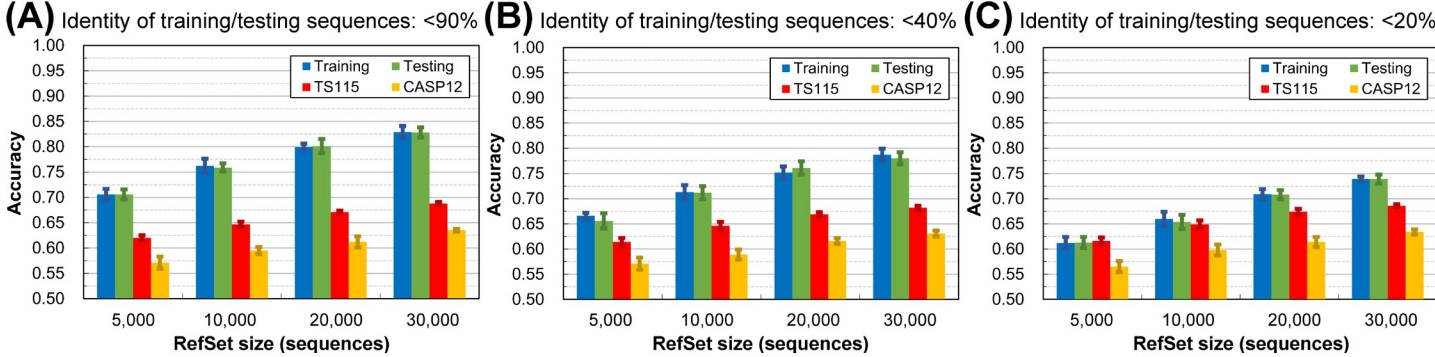

**Fig 14. The influence of reference dataset size on the accuracy of SSP.** (**A**) SSP accuracies of different reference dataset sizes under 90% identity of query sequences. (**B**) SSP accuracies of different reference dataset sizes under 40% identity of query sequences. (**C**) SSP accuracies of different reference dataset sizes under 20% identity of query sequences. The accuracy in training, testing, and independent tests all increased as the reference dataset expanded. Regardless of the homology of training/testing query sequences, the increasing trends of independent tests were nearly the same; meanwhile, the extent of accuracy increase in training and testing was greatly affected by the identity of query sequences. The gap of accuracy between training/testing and independent tests stands for over-fitting predictions. The overfitting in SSP will increase as the reference dataset size grows, especially when the homology of training/testing sequences is high.

CPUs were applied (disk space: 5.1GB). There were many experiments in this study, and most experiments were repeated more than ten times. For efficiency, we decided to use the conventional PSI-BLAST PSSM in most experiments.

## Determination of the appropriate size of the PSSM reference dataset for this study

The reference dataset size we used was much smaller than that used in most SSP works, resulting in lower accuracy than most published recently; therefore, we would like to examine the influences of reference dataset size on SSP accuracy and choose a suitable size for experiments in this work. Fig 13 illustrated the layout of this experiment, in which four reference dataset sizes were tested under three identity levels of training/testing query sequences. As expected, the accuracy increased as the reference dataset became larger (Figs 14 and S15). Very interestingly, regardless of whether the identity between query datasets was high (90%), medium (40%), or low (20%), the accuracies of independent tests increased with the same degree as the reference dataset expanded. In contrast, the increase in training and testing accuracy was much more significant at higher sequence identity than at lower identity. Thus, the gap between the accuracy of training/testing and independent tests was much larger for higher identity, indicating severer overfitting. These results reveal that the overfitting problem will become increasingly serious as the reference dataset expands, especially when the homology of training/testing sequences is high. In other words, reducing the homology of the training/testing query sequences is essential for preventing the overfitting of a developing SSP method, especially when the reference dataset is large. Considering that the NrPdb90-2015 (the source of reference datasets in most experiments) comprised only ~3,5000 proteins and in one of our experiments the reference dataset had to be divided into three subsets, we decided to use 10,000 proteins as the standard size of reference dataset in this study.

## Supporting information

**S1 Table. The version and parameter settings of the PSI-BLAST engine for the utilized SSP algorithms.**
(PDF)

**S2 Table. Performance verification of the applied state-of-the-art SSP algorithms.**
(PDF)

**S1 Fig. Effects of the homology between the training and testing query datasets on the SOV3 of secondary structure prediction.**
(TIF)

**S2 Fig. Effects of the homology within the training/testing query dataset on the SOV3 of secondary structure prediction.**
(TIF)

**S3 Fig. Effects of the isolation of PSSM reference sequences for training and evaluations on the SOV3 of secondary structure prediction.**
(TIF)

**S4 Fig. Effects of the homology between the query and PSSM reference datasets on the SOV3 of secondary structure prediction.**
(TIF)

**S5 Fig. Effects of the homology reduction of the PSSM reference dataset on the SOV3 of secondary structure prediction.**
(TIF)

**S6 Fig. The SOV accuracies of seven state-of-the-art secondary structure prediction methods evaluated using homology-reduced PSSM reference datasets.**
(TIF)

**S7 Fig. The SOV3 data of the feasibility tests for the proposed development and evaluation strategy for SSP methods.**
(TIF)

**S8 Fig. Effects of the homology between training and testing query datasets on the SOV3 of SSP obtained with decision tree or support vector machine.**
(TIF)

**S9 Fig. Verification of the effect of the homology between the query and PSSM reference datasets on the Q3 and SOV3 accuracy of SSP.**
(TIF)

**S10 Fig. Verification of the effect of the isolation of PSSM reference sequences on the development and evaluation of an SSP method.**
(TIF)

**S11 Fig. The SOV3 accuracy of machine learning algorithms obtained with training and testing datasets sharing decreasing levels of sequence homology.**
(TIF)

**S12 Fig. The SOV3 accuracy of combined machine learning algorithms obtained with training and testing datasets sharing decreasing levels of sequence homology.**
(TIF)

**S13 Fig. The Q8 accuracy achieved by a random forest of 60 artificial neural networks obtained with training and testing sets that share decreasing levels of sequence identities.**
(TIF)

**S14 Fig. The accuracy achieved by a random forest of 60 artificial neural networks obtained with HHBlits-PSSM and training and testing sets that share decreasing levels of sequence identities.**
(TIF)

**S15 Fig. The influence of reference dataset size on the SOV3 accuracy of SSP.**
(TIF)

**S1 File. The TS416 dataset.**
(FASTA)

## Acknowledgments

We would like to thank Yu-Wei Huang and Yen-Cheng Lin, students of WCL, for setting up the server machines and software packages. The progress of this project was greatly accelerated owing to the computing power offered by Prof. Jinn-Moon Yang and Jenn-Kang Hwang at National Yang Ming Chiao Tung University and Prof. Ping-Chiang Lyu at National Tsing Hua University, Taiwan.

## Author Contributions

**Conceptualization:** Chia-Hua Lo, Wei-Cheng Lo.

**Data curation:** Teng-Ruei Chen, Chia-Hua Lo, Sheng-Hung Juan, Wei-Cheng Lo.

**Funding acquisition:** Wei-Cheng Lo.

**Investigation:** Teng-Ruei Chen, Chia-Hua Lo, Wei-Cheng Lo.

**Methodology:** Teng-Ruei Chen, Chia-Hua Lo, Wei-Cheng Lo.

**Project administration:** Wei-Cheng Lo.

**Resources:** Wei-Cheng Lo.

**Software:** Teng-Ruei Chen, Sheng-Hung Juan, Wei-Cheng Lo.

**Supervision:** Wei-Cheng Lo.

**Validation:** Teng-Ruei Chen, Chia-Hua Lo, Sheng-Hung Juan, Wei-Cheng Lo.

**Writing – original draft:** Teng-Ruei Chen, Chia-Hua Lo, Sheng-Hung Juan, Wei-Cheng Lo.

**Writing – review & editing:** Chia-Hua Lo, Sheng-Hung Juan, Wei-Cheng Lo.

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
