## [Decision Letter · Decision Letter 0]

30 Jun 2020

PONE-D-20-11039

The influence of dataset homology and a rigorous evaluation strategy on protein secondary structure prediction

PLOS ONE

Dear Dr. Lo,

Thank you for submitting your manuscript to PLOS ONE. After careful consideration, we feel that it has merit but does not fully meet PLOS ONE’s publication criteria as it currently stands. Therefore, we invite you to submit a revised version of the manuscript that addresses the points raised during the review process.

Two experts in the field particularly appreciated the questions asked in your work. However, they noted significant shortcomings in the evaluation of the methods and must be answered. In addition, there is a great need to improve the text and its presentation, which are not of a professional level.

We look forward to receiving your revised manuscript.

Kind regards,

Alexandre G. de Brevern, Ph.D.

Academic Editor

PLOS ONE

Journal Requirements:

Reviewers' comments:

Reviewer's Responses to Questions

**Comments to the Author**

1. Is the manuscript technically sound, and do the data support the conclusions?

Reviewer #1: Yes

Reviewer #2: Partly

2. Has the statistical analysis been performed appropriately and rigorously? 

Reviewer #1: Yes

Reviewer #2: Yes

3. Have the authors made all data underlying the findings in their manuscript fully available?

Reviewer #1: Yes

Reviewer #2: Yes

4. Is the manuscript presented in an intelligible fashion and written in standard English?

Reviewer #1: No

Reviewer #2: Yes

5. Review Comments to the Author

Reviewer #1: This paper studied the impact of homologies between and within datasets is evaluating SS prediction performances of various prediction methods. A major result is the discovery that the sequence homology redundancy between or within most datasets would make the accuracy of an SSP algorithm overestimated while the redundancy within the target dataset would make the accuracy underestimated. Although this conclusion is quite obvious to researchers in the field, the detailed experiments and empirical results to support and quantify the impact have some value.

The papers studied the impact using some relatively older prediction methods. The impact on more recent deep learning based tools, such as MUFOLD_SS, SPOT-1D, NetSurfP2, Porter5, could be reported in the revision to make the paper stronger.

The paper proposed a new method to perform more rigorous and objective performance evaluation. However, the result of the new method seems to be very similar to previous method in some cases, e.g., Q3 average changes from 0.810 to 0.807. Using the proposed method on state-of-the-art SS prediction tools, such as MUFOLD_SS, SPOT-1D, NetSurfP2, Porter5, to generate a comprehensive evaluation result in the revision will be a valuable contribution to the field and can be used by other researchers in the field as baselines for future development.

The paper is not very well written and is confusing at places because of using terms not commonly used or with different meanings in the field. For example, the paper uses "target" and "query" to represent the "database" and "input sequence set". However, in this field "target/query" usually means "input sequence". The author should use commonly use terms or clearly define the meanings of the terms they use.

Reviewer #2: The paper asked the question of how choosing the sequence identity levels in the target sequence set, the training set, and the test set would affect the generalization performance of a trained secondary structure prediction model on unseen proteins. The two independent test sets of proteins used in the experiment were the TS115 and CASP 12. The authors designed experiments to answer the questions raised, then based on the results proposed a development and evaluation protocol to train a secondary structure prediction model.

The questions asked in the paper are meaningful, and the experiments in the paper are mostly well designed. However the two independent test sets used in the paper are from 2016, while many new protein structures have been released by PDB since 2016 (https://www.rcsb.org/stats/growth/growth-released-structures

). I believe the conclusions of the experiments will be persuasive if the trained models in the paper are tested on a new and large independent test set. I propose the authors to expand their experiments to validate their findings:

1) Major step: Build a new independent test set with the new protein structures released in PDB since 2016 until now (e.g. using the same protocol that is used to build the TS115 testset), and test all the trained secondary structure prediction models on the new independent test set to validate all the paper findings.

2) As HHBlits (https://www.nature.com/articles/nmeth.1818) has become a popular tool do determine homologous protein sequences from a target sequence set for a query protein (table 1 of https://www.sciencedirect.com/science/article/pii/S2001037019304441 ), the authors should validate all their findings when HHBlits is used to build the PSSM for each query protein sequence.

6. PLOS authors have the option to publish the peer review history of their article (what does this mean?). If published, this will include your full peer review and any attached files.

Reviewer #1: No

Reviewer #2: No

---

## [Author Response · Author response to Decision Letter 0]

1 Mar 2021

PONE-D-20-11039

The influence of dataset homology and a rigorous evaluation strategy on protein secondary structure prediction

Dear Professors and Editor,

We would like to thank the anonymous reviewers for their careful reading and detailed comments, which have greatly enhanced this article. We are pleased that you find sufficient merit in this work as to ask for an appropriately revised manuscript. We also feel very sorry for keeping you waiting so long. The experiments of this study were all time and computation costly. The new version of our manuscript has been modified according to referees' comments, which are also answered as follows.

Comments to the Author

1. Is the manuscript technically sound, and do the data support the conclusions?

Reviewer #1: Yes

Reviewer #2: Partly

Response:

We would like to thank the reviewers for their constructive comments. Experiments on additional independent datasets and new protein secondary structure prediction (SSP) algorithms have been performed to make this work stronger.

2. Has the statistical analysis been performed appropriately and rigorously?

Reviewer #1: Yes

Reviewer #2: Yes

Response:

We would like to thank the reviewers for their positive comments.

3. Have the authors made all data underlying the findings in their manuscript fully available?

Reviewer #1: Yes

Reviewer #2: Yes

Response:

We thank the reviewers for their time and patience in reviewing our numerous figures supporting data files.

4. Is the manuscript presented in an intelligible fashion and written in standard English?

Reviewer #1: No

Reviewer #2: Yes

Response:

We thank the reviewers for the comments. Please see below for our response to the English writing of the revised manuscript.

5. Review Comments to the Author

----------

----------

Reviewer #1: This paper studied the impact of homologies between and within datasets is evaluating SS prediction performances of various prediction methods. A major result is the discovery that the sequence homology redundancy between or within most datasets would make the accuracy of an SSP algorithm overestimated while the redundancy within the target dataset would make the accuracy underestimated. Although this conclusion is quite obvious to researchers in the field, the detailed experiments and empirical results to support and quantify the impact have some value.

1) The papers studied the impact using some relatively older prediction methods. The impact on more recent deep learning based tools, such as MUFOLD_SS, SPOT-1D, NetSurfP2, Porter5, could be reported in the revision to make the paper stronger.

Response:

We sincerely thank the reviewer for the constructive comment and valuable information, which greatly help us improve this work. Among the recommended tools, we have applied MUFOLD_SS, NetSurfP2, and Porter5 to perform large-scale assessments on the proposed SSP evaluation strategy. SPOT-1D was also installed. However, predicting one protein of around 200 residues with this tool using our rapidest server machine (equipped with 768G RAM, 16-thread Intel i9 5.1GHz CPU, and Nvidia GTX 1280-core 1.7GHz GPU) took 9.6 days. Due to the limited time and resources, we changed to use Spider3, another algorithm published a little earlier than SPOT-1D by the same research team. In fact, the SPOT-1D makes SSP by integrating the prediction results of Spider3 and the PSSMs generated by PSI-BLAST and HHBlits.

In the subsection "The proposed development and evaluation strategy for SSP methods" of Results, we described how these new algorithms were applied, and the experimental results are summarized on Pages 22–24 and in Fig 7. The rise-and-drop trends and overestimation of SSP accuracy we observed based on our in-house predictor and the algorithms published before 2016 were verified using these up-to-date tools.

The SSP algorithms we previously utilized were all published before 2016 because this study was started in 2016. Owing to the multiple repeats for each test group and the long computing time of applied algorithms, every experiment was very time-consuming. It took us three years to finish the first manuscript. When this project was started, most SSP algorithms, including the seven we assessed, used PSI-BLAST to generate PSSM as the SSP feature set. In recent years, state-of-the-art algorithms work based on PSSMs generated by both PSI-BLAST and HHBlits (e.g., MUFOLD_SS, Porter5, and Spider3), or only HHBlits (NetSurfP2). The fact that different types of algorithms showed the same accuracy decay in the face of experimental datasets with decreasing homology supported our concern about the over-evaluation of SSP. We are genuinely looking forward that the proposed SSP development and evaluation strategy can be applied in future SSP works. Hopefully, the robustness of the new algorithms developed based on it can help move forward research and industrial fields depending on SSP.

In addition to Results, the methods recommended by the reviewer were mentioned in several places in the article. We have marked them with a light green text background.

----------

2) The paper proposed a new method to perform more rigorous and objective performance evaluation. However, the result of the new method seems to be very similar to previous method in some cases, e.g., Q3 average changes from 0.810 to 0.807. Using the proposed method on state-of-the-art SS prediction tools, such as MUFOLD_SS, SPOT-1D, NetSurfP2, Porter5, to generate a comprehensive evaluation result in the revision will be a valuable contribution to the field and can be used by other researchers in the field as baselines for future development.

Response:

Thank you very much for this comment, which let us know how misleading we had described the data. The Q3 average changed from 0.810 to 0.807 shown in the original Table 2 in Materials and Methods was not because of the proposed strategy. In that table, some accuracies were even improved (e.g., the Q3 average of CASP12 changed from 0.791 to 0.800). This study utilized many SSP algorithms, and they used different versions and settings of PSI-BLAST to generate PSSM (see S1 Table). In order to make fair assessments, we needed to unify the version and settings of PSI-BLAST used by the script programs of those algorithms. However, we were afraid that the performance of some algorithms might be ruined due to modified settings. Therefore, before those algorithms were actually applied in any experiment, we checked whether they performed normally with the unified settings.

In the previous manuscript, we said before Table 2 that we had "preliminarily tested" the accuracy and used the term "performance pretest" to start the title of Table 2. We now realize that these descriptions were misleading and have changed to "preliminarily measured" and "preliminary verifications". The statements about PSI-BLAST usage were carefully revised (see the sky blue text on Pages 44–46). To prevent distracting the reader, we have also moved the original Table 2 to the S2 Table of the supporting information files (Page 62).

----------

3) The paper is not very well written and is confusing at places because of using terms not commonly used or with different meanings in the field. For example, the paper uses "target" and "query" to represent the "database" and "input sequence set". However, in this field "target/query" usually means "input sequence". The author should use commonly use terms or clearly define the meanings of the terms they use.

Response:

We would like to thank the reviewer for pointing out these shortcomings. The definitions of "target" and "query" were made, not clearly enough, in Introduction and Fig 1. We know that a "target" sequence means very differently in SSP and the field of sequence similarity search. In the revised article, we stop using "target", and the dataset used to generate PSSM by PSI-BLAST or HHBlits is now called the "reference" dataset. The definitions of reference, query, and independent test datasets are made in the new Introduction on Page 5. In figures, the reference dataset is abbreviated as "RefSet".

Besides, we add some summary statements to make the main findings easier for readers to catch. For instance, most SSP works emphasized low homology between the training and independent test query datasets. A highly redundant PSSM reference dataset, e.g., UniRef90, is commonly applied to develop algorithms, and the homology between the reference dataset and the training and independent datasets was not manipulated. We discovered that the homology between the training and independent query datasets exerts little effect on SSP. Contrarily, the homology within the reference dataset and that between reference and query datasets greatly influence the reliability of SSP evaluations.

The old manuscript contained many redundancies partly because we took PLOS ONE's suggestion of the figure legend too seriously: "allow readers to understand it without referring to the text". We have removed many redundant statements from the figure legends and simplified the main text.

To improve this article's quality, we have carefully refined every paragraph and corrected grammatical/spelling errors. Comparing the new manuscript with the previous one, almost every subsection is extensively revised. The tracking of Microsoft Word was retained in the uploaded "Revised Manuscript with Track Changes" file. We hope these changes will meet the criteria for publication in PLOS ONE.

----------

----------

Reviewer #2: The paper asked the question of how choosing the sequence identity levels in the target sequence set, the training set, and the test set would affect the generalization performance of a trained secondary structure prediction model on unseen proteins. The two independent test sets of proteins used in the experiment were the TS115 and CASP 12. The authors designed experiments to answer the questions raised, then based on the results proposed a development and evaluation protocol to train a secondary structure prediction model.

The questions asked in the paper are meaningful, and the experiments in the paper are mostly well designed. However the two independent test sets used in the paper are from 2016, while many new protein structures have been released by PDB since 2016 (https://www.rcsb.org/stats/growth/growth-released-structures). I believe the conclusions of the experiments will be persuasive if the trained models in the paper are tested on a new and large independent test set. I propose the authors to expand their experiments to validate their findings:

1) Major step: Build a new independent test set with the new protein structures released in PDB since 2016 until now (e.g. using the same protocol that is used to build the TS115 testset), and test all the trained secondary structure prediction models on the new independent test set to validate all the paper findings.

Response:

We would like to thank the reviewer for the positive feedback and helpful suggestion, which have provided more robust supports to the conclusions of this study. First, we utilized a newer CASP dataset, i.e., the CASP13, which contained novel PDB proteins deposited mainly between 2017 and 2018. Besides, using the protocol Dr. Yaoqi Zhou's team built the TS115 dataset (115 proteins) [44], we established a new independent test set named TS416. This dataset consisted of 416 proteins deposited in the PDB after Jan. 2019 and highly non-redundant from proteins deposited before Dec. 2018 (see S1 File). To ensure its non-redundancy from PDB 2018, when using PSI-CD-HIT-2D to establish the TS416, we applied a 25% sequence identity cutoff, lower than the 30% cutoff applied to TS115 in [44]. Besides, the internal homology of the TS416 was also made low by performing iterative homology reductions using CD-HIT, USEARCH, and MMseqs2 until no more sequence could be eliminated with a 25% identity cutoff.

Adding CASP13 and TS416, we repeated the experiments and found that the accuracies obtained with them were close to those obtained with CASP12, and the trends of the accuracies of these three datasets were the same (see Figs 11 and 12, and S14 Fig). Because of the proximity of curves drawn according to these datasets and the severe overlapping of error bars, we only show the results of CASP13 and TS416 in the experiments described in Materials and Methods and the final large-scale experiment described in Results. In the large-scale experiment, the trends of accuracy obtained with CASP13 and TS416 using state-of-the-art SSP algorithms were also very similar to those obtained with CASP12 and TS115 (see Fig 7 for Q3/8, and S7 Fig for SOV3/8).

No matter using independent test datasets of PDB proteins deposited in what years (before 2017: CASP12 and TS115; after 2017: CASP13 and TS416), the trends of accuracy were all highly similar. This fact has strengthened the conclusions we made in the previous manuscript. In the revised manuscript, we marked the contents about CASP13 and TS416 with an orange text background except some parts already marked with another color.

----------

2) As HHBlits (https://www.nature.com/articles/nmeth.1818) has become a popular tool to determine homologous protein sequences from a target sequence set for a query protein (table 1 of https://www.sciencedirect.com/science/article/pii/S2001037019304441), the authors should validate all their findings when HHBlits is used to build the PSSM for each query protein sequence.

Response:

We thank the reviewer for the suggestion. Before receiving this pee-review report, we had known that HHBlits is increasingly applied in SSP and felt very interested. We have accordingly installed HHBlits v3.3.0 and tested its feasibility. Prior to homology searches by either PSI-BLAST or HHBlits, the target dataset (or reference dataset) must be formatted into a specialized format. The formatting procedure of PSI-BLAST is very efficient. With only one CPU, it typically took <0.5 sec for 10,000 proteins. The produced database files used ~3.2MB disk space. Very differently, because of the sophisticated multiple sequence alignment analyses conducted by HHBlits, formatting a 10,000-protein dataset by it cost approximately 1.3 days when 12 Intel Xeon 3.33GHz CPUs were applied. The produced database files needed 5.1GB disk space. For efficiency, we kept using the PSI-BLAST PSSM in most experiments. The feasibility of HHBlits PSSM was tested in two experiments.

In Materials and Methods, after the machine learning framework was determined, a new subsection entitled "Determination of the type of PSSM for experiments" described how we tested the feasibility of HHBlits PSSM. Results diagramed in the new S14 Fig indicated that, based on 10,000-protein reference datasets, the HHBlits PSSM remarkably improved the SSP accuracy of our in-house predictor by ~2.5% when compared with the PSI-BLAST PSSM; meanwhile, the trends of HHBlits PSSM's accuracy in training, testing, and independent tests were similar to those of PSI-BLAST PSSM. This small-scale experiment was repeated by random sampling 10 times, which totally took 146 days using the second rapidest server machine of ours (12 Intel Xeon 3.33GHz CPUs and 166GB RAM).

In Results, we performed a large-scale experiment using reference datasets of one million sequences. In addition to the seven state-of-the-art SSP algorithms assessed previously (they all worked based on PSI-BLAST PSSM), four up-to-date SSP algorithms were also applied in that experiment. Among the four new algorithms, MUFOLD_SS, Porter5, and Spider3 made predictions based on PSSMs generated by both PSI-BLAST and HHBlits, and NetSurfP2 only utilized HHBlits. As shown in the revised Fig 7 and S7 Fig, when assessed with CASP13 and TS416 independent test datasets, the average accuracy of these HHBlits-powered algorithms was around 3.5% higher than that of PSI-BLAST PSSM-based algorithms. Nevertheless, the trends of accuracy obtained by both types of algorithms were very similar. The observed accuracy of all algorithms went highest when the sequence identities of experimental datasets fell between 90% and 80%, and the lowest accuracy occurred at identities ≤50%. In addition to dataset formatting, the newly applied SSP algorithms' prediction processes were also time costly. This large-scale experiment was repeated three times. The computation was distributed to three server machines but still cost more than two months.

Since the same trends of accuracy were obtained in both small and large scale tests by algorithms using either PSI-BLAST or HHBlits PSSM, the conclusions of this work are supposed sufficiently substantial to form the foundation of future SSP developments. In the revised manuscript, the new contents about HHBlits and HHBlits-powered SSP algorithms are marked with a light green text background.

6. Do you want your identity to be public for this peer review?

Reviewer #1: No

Reviewer #2: No

Response:

We would like to express again our sincere thanks to the anonymous reviewers for helping us improve this work.

---

## [Decision Letter · Decision Letter 1]

29 Jun 2021

The influence of dataset homology and a rigorous evaluation strategy on protein secondary structure prediction

PONE-D-20-11039R1

Dear Dr. Lo,

We’re pleased to inform you that your manuscript has been judged scientifically suitable for publication and will be formally accepted for publication once it meets all outstanding technical requirements.

Kind regards,

Alexandre G. de Brevern, Ph.D.

Academic Editor

PLOS ONE

Additional Editor Comments (optional):

Reviewers' comments:

Reviewer's Responses to Questions

**Comments to the Author**

1. If the authors have adequately addressed your comments raised in a previous round of review and you feel that this manuscript is now acceptable for publication, you may indicate that here to bypass the “Comments to the Author” section, enter your conflict of interest statement in the “Confidential to Editor” section, and submit your "Accept" recommendation.

Reviewer #2: All comments have been addressed

Reviewer #3: All comments have been addressed

2. Is the manuscript technically sound, and do the data support the conclusions?

Reviewer #2: Yes

Reviewer #3: Yes

3. Has the statistical analysis been performed appropriately and rigorously? 

Reviewer #2: Yes

Reviewer #3: Yes

4. Have the authors made all data underlying the findings in their manuscript fully available?

Reviewer #2: Yes

Reviewer #3: Yes

5. Is the manuscript presented in an intelligible fashion and written in standard English?

Reviewer #2: Yes

Reviewer #3: No

6. Review Comments to the Author

Reviewer #2: The authors of the manuscript have done many more experiments in the revision according to the reviewers feedback. The content of the paper is now up to date, and I am happy to see it published.

Reviewer #3: This paper aims at reassess impact of sequence homology in datasets used for SSP. They have done quite rigorous study and have addressed all the questions that were raised by the reviewers in the first review. However, I would request them to go thoroughly review how they have written certain terms figure labels. There is clearly overuse of the article "The". Although, I'm able to understand what the authors want to convey, it is not easy always. Hence, my suggestion to reedit their manuscript so that any reader can follow their work with ease.

7. PLOS authors have the option to publish the peer review history of their article (what does this mean?). If published, this will include your full peer review and any attached files.

Reviewer #2: No

Reviewer #3: No

---

## [Editor Report · Acceptance letter]

5 Jul 2021

PONE-D-20-11039R1 

The influence of dataset homology and a rigorous evaluation strategy on protein secondary structure prediction 

Dear Dr. Lo:

I'm pleased to inform you that your manuscript has been deemed suitable for publication in PLOS ONE. Congratulations! Your manuscript is now with our production department. 

Kind regards, 

on behalf of

Dr. Alexandre G. de Brevern 

Academic Editor

PLOS ONE